# Chemistry-driven changes strongly influence climate forcing from vegetation emissions

James Weber [1,5] ✉, Scott Archer-Nicholls[1,6], Nathan Luke Abraham[1,2], Youngsub Matthew Shin[1], Paul Griffiths [1,2], Daniel P. Grosvenor[3], Catherine E. Scott [4] & Alex T. Archibald[1,2]

Biogenic volatile organic compounds (BVOCs) affect climate via changes to aerosols, aerosol-cloud interactions (ACI), ozone and methane. BVOCs exhibit dependence on climate (causing a feedback) and land use but there remains uncertainty in their net climatic impact. One factor is the description of BVOC chemistry. Here, using the earth-system model UKESM1, we quantify chemistry's influence by comparing the response to doubling BVOC emissions in the pre-industrial with standard and state-of-science chemistry. The net forcing (feedback) is positive: ozone and methane increases and ACI changes outweigh enhanced aerosol scattering. Contrary to prior studies, the ACI response is driven by cloud droplet number concentration (CDNC) reductions from suppression of gas-phase $SO_2$ oxidation. With state-of-science chemistry the feedback is 43% smaller as lower oxidant depletion yields smaller methane increases and CDNC decreases. This illustrates chemistry's significant influence on BVOC's climatic impact and the more complex pathways by which BVOCs influence climate than currently recognised.

Atmospheric composition, and its response to a perturbation, plays a key role in climate[1]. Tropospheric chemistry in current state-of-the-art climate models used in the 6th Coupled Model Intercomparison Project (CMIP6) is highly parameterised in terms of reactions, emissions, aerosol chemistry and gas-aerosol coupling and there remains considerable uncertainty in the modelling of chemistry in the lower atmosphere.

In a climatic context this uncertainty is important because tropospheric chemistry is a major factor in determining the atmosphere's oxidative capacity. Oxidants control the lifetimes of methane ($CH_4$), and thus its efficacy as a greenhouse gas (GHG), and a huge range of reactive gases, including volatile organic compounds (VOCs). Oxidation of VOCs in the presence of nitrogen oxides can produce ozone ($O_3$), another GHG. Unlike $CH_4$, well-mixed in the troposphere, $O_3$ is spatially heterogeneous. $O_3$'s potency as a GHG is much greater in the cold upper troposphere[2] and thus dependent on dispersion of $O_3$-precursors. Oxidants also influence aerosol processes, termed aerosol-oxidant coupling, through the oxidation of sulfur dioxide ($SO_2$) to sulfate aerosol and of VOCs to low volatility species which can contribute to secondary organic aerosol (SOA). Aerosols influence climate directly by scattering or absorbing solar radiation and indirectly by affecting cloud properties[3]. Oxidants control where the key reactions for aerosol production occur and therefore influence the resulting aerosol's lifetime, effect on cloud properties, and consequently their climatic impact[4,5].

Biogenic volatile organic compounds (BVOCs) play a central role in these chemistry-climate interactions by influencing oxidant concentrations, via direct reaction and secondary production from oxidation products[6], and providing condensable material for SOA (e.g.,[7]). However, BVOC emissions ($E_{BVOC}$) depend strongly on climate

---

[1]Centre for Atmospheric Science, Yusuf Hamied Department of Chemistry, University of Cambridge, Cambridge CB2 1EW, UK. [2]National Centre for Atmospheric Science, Yusuf Hamied Department of Chemistry, University of Cambridge, Cambridge CB2 1EW, UK. [3]Institute for Climate and Atmospheric Science, School of Earth and Environment, University of Leeds, Leeds LS2 9JT, UK. [4]School of Earth and Environment, University of Leeds, Leeds LS2 9JT, UK. [5]Present address: School of Biosciences, University of Sheffield, Sheffield S10 2TN, UK. [6]Present address: Research IT, University of Manchester, Manchester M13 9PL, UK. ✉e-mail: j.weber@sheffield.ac.uk

themselves, leading to a BVOC-climate feedback (BCF). Determining the sign and magnitude of this feedback is important for predicting future climate change (e.g.,[8]).

$E_{BVOC}$, especially isoprene (the most widely emitted BVOC[9]), are strongly dependent on atmospheric conditions and land use. Rising $CO_2$ inhibits isoprene production[10] but also drives increased vegetation-mass via fertilisation[11]. Higher temperatures also increase emissions of isoprene and monoterpenes[12]. Perturbations to aerosols and clouds change photosynthetically active radiation (PAR) and precipitation, also influencing emissions[13,14]. Simulated isoprene emissions exhibit increases from the present day to 2100, albeit with significant variation between both models and future climate scenarios[15]. Proposed re/afforestation policies would likely drive even greater increases in BVOC emissions.

The climatic impact of BVOCs has been studied with varying degrees of sophistication over the last two decades. Most studies predict that increases to SOA following enhanced $E_{BVOC}$ would cause a negative radiative forcing (RF) (via increased aerosol scattering and cloud albedo (e.g.,[16,17])), constituting a negative feedback. When changes to gas phase chemistry are considered, increases to $CH_4$ lifetime and $O_3$ cause a positive RF, although the extent to which this opposes the negative RF from aerosols is uncertain. Ref. 18 found the negative RF from aerosols still outweighed the positive forcing from $O_3$ and $CH_4$ while[19], using a different model, found the opposite. AerChemMIP also revealed significant inter-model variation in the response to $2\times E_{BVOC}$ (Fig. S1) with UKESM1 and GISS predicting a positive forcing while GFDL and CESM2 a negative forcing[20]. The impact of oxidant changes on sulfate aerosol from increased $E_{BVOC}$ has not previously been examined in detail and is a key factor in this work.

Thus, the uncertainty in BVOCs' climatic impact depends on the uncertainty in multiple chemical and physical processes governing the net radiative forcing. Several studies have investigated how modelling aerosol processes (principally nucleation, condensation and growth) can affect BVOCs' climatic impact[21–23]. By contrast, there has been no rigorous assessment of the influence the description of BVOCs' chemistry, and the effects to oxidants, has on the climatic impact of BVOCs despite recent advancements in the understanding of this chemistry[24,25]. For isoprene this centres on reactions of the peroxy radical formed by reaction with OH (ISOPOO) (Fig. 1). Some ISOPOO isomers can undergo intramolecular hydrogen shifts (H-shifts) which produce species which regenerate OH, termed $HO_x$-recycling. These reactions, along with natural emissions of $NO_x$ from soil, increase simulated OH in environments with high isoprene emissions and low anthropogenic and biomass burning emissions of $NO_x$, helping to reconcile the persistent model low biases for OH against observations[26,27]. We expect the smaller depletion of OH by isoprene with this chemistry to have ramifications for the atmospheric and radiative response to an $E_{BVOC}$ perturbation via changes to $CH_4$, $O_3$, aerosol and cloud properties.

In this study we assess how the description of BVOC chemistry affects the simulated climatic impact of BVOCs. We compare the change to the atmosphere's composition and energy balance, specifically the RF, following a doubling of $E_{BVOC}$ in the preindustrial atmosphere (PI) with two chemical mechanisms. We use Strat-Trop (ST)[28], the standard mechanism in UKESM1 and practical for long climate studies, and CRI-Strat 2 (CS2)[29]. CS2 includes a much more comprehensive description of tropospheric chemistry including ISOPOO H-shifts and a more complete treatment of monoterpene oxidation,

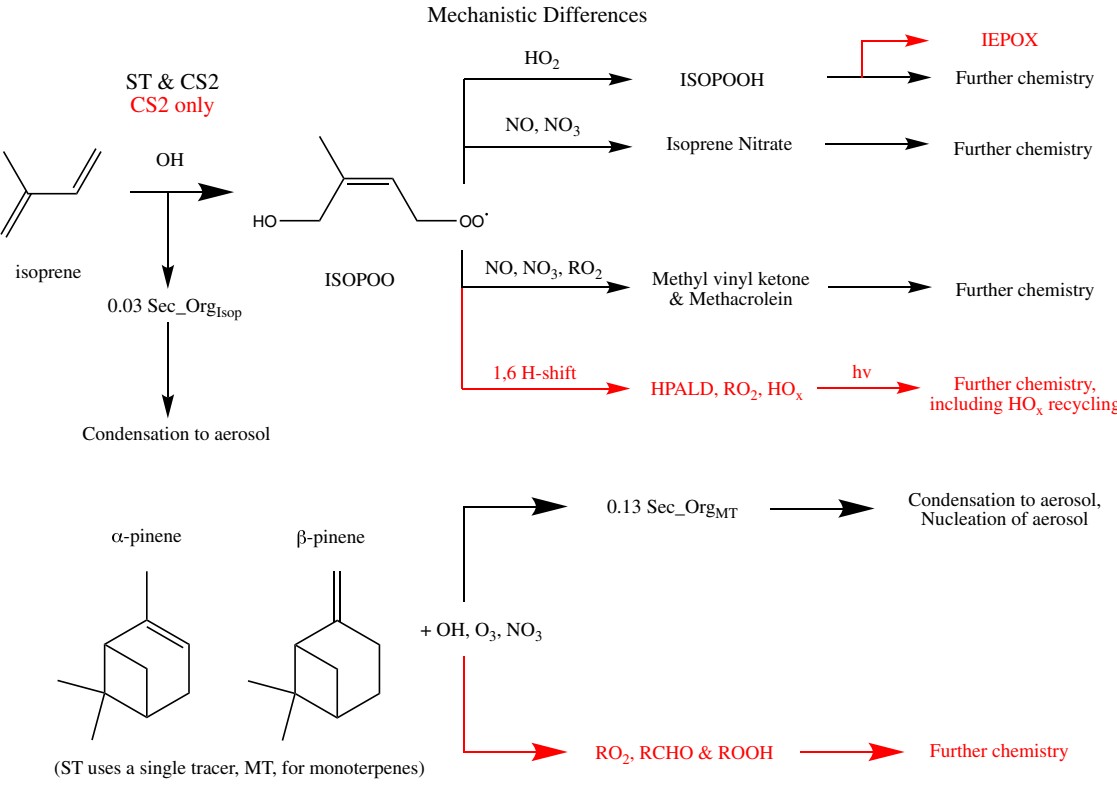

**Fig. 1 | Mechanistic differences between Strat-Trop (ST) and CRI-Strat 2 (CS2) for the key processes of isoprene oxidation by OH and oxidation of monoterpenes (represented by _α_-pinene and _β_-pinene in CS2).** Processes in black are featured in ST and CS2 while processes in red are only in CS2. $RO_2$, RCHO and ROOH refer to peroxy radicals, carbonyl and hydroperoxides respectively.

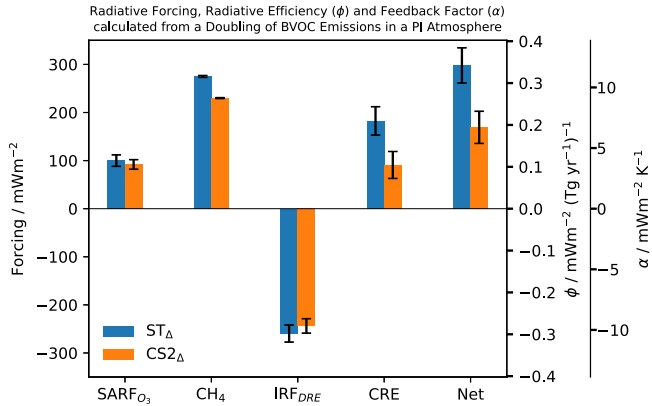

**Fig. 2 | Radiative forcing, radiative efficiency ($\phi$) and feedback factor ($\alpha$) from a doubling of BVOC emissions in a pre-industrial (PI) atmosphere.** We show the individual forcing components from changes to $O_3$ (SARF$_{O3}$), CH$_4$, the aerosol direct radiative effect (IRF$_{DRE}$) and aerosol-cloud interactions (CRE) and their combined totals (Net) for Strat-Trop (ST$_\Delta$) and CRI-Strat 2 (CS2$_\Delta$). The left axis shows the radiative forcing, the inner right axis the radiative efficiency and the outer right axis the feedback factor. Error bars show the standard error.

with both important for oxidant production. These differences are described in Fig. 1 and Methods. Following 2xE$_{BVOC}$ we find the positive RF from changes to $O_3$, CH$_4$ and ACI outweighs the negative RF from aerosol scattering but the net RF is 43% smaller with CS2 due to a smaller depletion in oxidants. This highlights the multiple pathways by which chemistry, oxidants and aerosols interact to affect radiatively-active atmospheric components and thus demonstrates the importance of uncertainty in BVOC chemistry.

## Results and discussion

Figure 2 shows the RF, radiative efficiency and feedback factor from changes in $O_3$ (SARF$_{O3}$), CH$_4$, aerosol scattering (IRF$_{DRE}$) and the interactions of clouds with radiation, termed the cloud radiative effect (CRE) (Methods). Mechanism acronyms ST or CS2 refer to a particular detail of the mechanism (e.g. the OH + CH$_4$ rate constant in ST). Individual runs are denoted with the mechanism acronym and subscript (e.g. ST$_{con}$, ST$_{2x}$ for the control run and run with doubled BVOC (2xE$_{BVOC}$) respectively). ST$_\Delta$ and CS2$_\Delta$ refer to the change between the control and 2xE$_{BVOC}$ simulations for a given parameter (e.g., the change in $O_3$ in ST$_\Delta$ refers to the change in $O_3$ for ST$_{2x}$ - ST$_{con}$) (Methods). Both mechanisms simulate a net positive radiative forcing (and therefore a positive feedback), but the forcing in CS2$_\Delta$ (168 ± 33 mWm$^{-2}$) is 43% smaller than ST$_\Delta$ (298 ± 37 mWm$^{-2}$). This is driven by smaller positive forcings from CH$_4$ (−45 mWm$^{-2}$; −16%) and CRE (−92 mWm$^{-2}$; −50%) in CS2$_\Delta$ compared to ST$_\Delta$ which, along with the 8% smaller SARF$_{O3}$ (−8 mWm$^{-2}$), outweigh the 7% (16 mWm$^{-2}$) stronger negative IRF$_{DRE}$ in ST$_\Delta$. The negative IRF$_{DRE}$ and positive CH$_4$ and SARF$_{O3}$ forcings following an E$_{BVOC}$ increase are qualitatively in agreement with prior studies (e.g.,[19,20]), but the positive CRE contrasts with most studies[18,23]: both simulated negative CRE with increased E$_{BVOC}$. The key processes controlling these forcings and the factors driving the mechanistic differences are now reviewed.

### The hydroxyl radical & methane

The larger positive CH$_4$ forcing in ST$_\Delta$ than CS2$_\Delta$ can be understood with reference to changes in the OH concentration. 2xE$_{BVOC}$ depletes OH throughout the troposphere in both ST$_{2X}$ and CS2$_{2X}$ but the larger relative reduction in ST$_\Delta$ of −31% (cf. −24% in CS2$_\Delta$) is one of the fundamental causes of the different climatic responses between the chemical mechanisms. In the lowest 5 km, OH decreases by >65% (>55%) and >50% (>35%) over Amazonia and central Africa respectively

in ST$_\Delta$ (CS2$_\Delta$), two of the regions with greatest BVOC emissions (Fig. 3a, b). In the lowest ~1 km CS2's enhanced HO$_x$-recycling from isoprene is particularly influential while in the lower tropical FT (~1–5 km) the mechanistic differences come from a greater increase in OH production from HO$_2$ + NO and hydroperoxide (ROOH) photolysis, primarily coming from the ROOH derived from $\alpha$-pinene and $\beta$-pinene (omitted in ST), in CS2$_\Delta$.

CS2 produces higher yields of the major hydroperoxide (H$_2$O$_2$) than ST from the ozonolysis of isoprene (38.5% vs. 9%). The consideration of monoterpene chemistry in CS2, in contrast to ST, also leads to higher production of H$_2$O$_2$ (18% direct yield vs. zero in ST) as well as HO$_2$-precursors (e.g., HCHO). Thus, 2xE$_{BVOC}$ produces a greater increase in H$_2$O$_2$ and HO$_2$ in CS2$_\Delta$ than ST$_\Delta$, driving a greater increase in secondary OH production in the lower FT (Fig. S2a, b).

As the major tropospheric sink for CH$_4$, the decrease in OH following 2xE$_{BVOC}$ leads to reductions in CH$_4$ oxidation and increases in simulated CH$_4$ concentration. The reduction in oxidation flux is greatest in the warm tropical lower troposphere (Fig. 3c) given the large OH reduction and strong positive temperature dependence of OH + CH$_4$. The larger reduction of OH in ST$_\Delta$ leads to a larger decrease in CH$_4$ oxidation flux (Fig. 3d), corresponding to larger increases in CH$_4$ concentration (ST$_\Delta$ 276 ppbv vs. CS2$_\Delta$ 223 ppbv) and forcing (ST$_\Delta$ 275 mWm$^{-2}$ vs. CS2$_\Delta$ 230 mWm$^{-2}$) (Methods).

### Ozone

The forcing from $O_3$ changes is dictated by the partitioning of nitrogen between reactive NO$_x$ and reservoir species (predominantly peroxyacetylnitrate (PAN) and nitric acid (HONO$_2$)), the availability of peroxy radical (RO$_2$) precursors and the location of $O_3$ production: the radiative efficiency of $O_3$ (forcing per unit change in concentration) is greater around the tropical tropopause than in the lower troposphere[2].

2xE$_{BVOC}$ reduces PBL $O_3$ over the major biogenic emission regions via $O_3$'s direct reaction with BVOCs. PAN formation also increases but mechanistic differences mean PAN has a ~35% longer lifetime in the warm PBL in ST than CS2. This leads to greater vertical transport of PAN into the FT where the lower temperature increases PAN's lifetime (from ~1 h in the PBL to ~2 days in the FT).

The increase in PAN in the middle troposphere in both mechanisms leads to lower NO$_x$ throughout the region and a reduction in $O_3$, greater in ST$_\Delta$. However, around the tropical tropopause, increases in HO$_2$, driven by the photolysis of carbonyls (RCHO) such as HCHO produced from BVOC oxidation products, result in increased $O_3$ via the reaction of HO$_2$ + NO and subsequent NO$_2$ photolysis. The increase in HO$_2$, and thus $O_3$, is greater in ST$_\Delta$ since the greater reduction of OH leads to greater vertical transport of these HO$_2$ precursors, allowing them to reach the region with maximum $O_3$ radiative efficiency. By contrast, HO$_2$ production from carbonyl photolysis increases by more in CS2$_\Delta$ in the lower and middle tropical troposphere where $O_3$'s radiative efficiency is lower (Fig. 3e, f).

The result is an 8% smaller forcing in CS2$_\Delta$ (92 ± 9 mWm$^{-2}$) compared to ST$_\Delta$ (100 ± 10 mWm$^{-2}$) despite CS2$_\Delta$ producing a 20% greater increase in tropospheric $O_3$ burden; highlighting the influence of $O_3$ precursor-transport and thus oxidant concentrations.

### Aerosol scattering (IRF$_{DRE}$)

The increase in E$_{BVOC}$ not only increases the fuel for SOA production (and thus burden), but also, via oxidant depletion, influences the location of SOA production. The reduction in OH (Fig. 3a, b) increases BVOC lifetimes meaning SOA-precursors are formed at higher altitude, further from E$_{BVOC}$ sources, and the resulting SOA has a longer lifetime and greater climatic impact. The greater OH reduction in ST$_\Delta$ yields greater increases in isoprene lifetime (8.2 h (66%) vs. 3.7 h (61%)) and thus transport away from source than in CS2$_\Delta$. Accordingly, SOA burden (lifetime) increases by 121% (12%) in ST$_\Delta$ compared to 114% (7%) in CS2$_\Delta$. The greater vertical transport of SOA-precursors in ST$_\Delta$ also

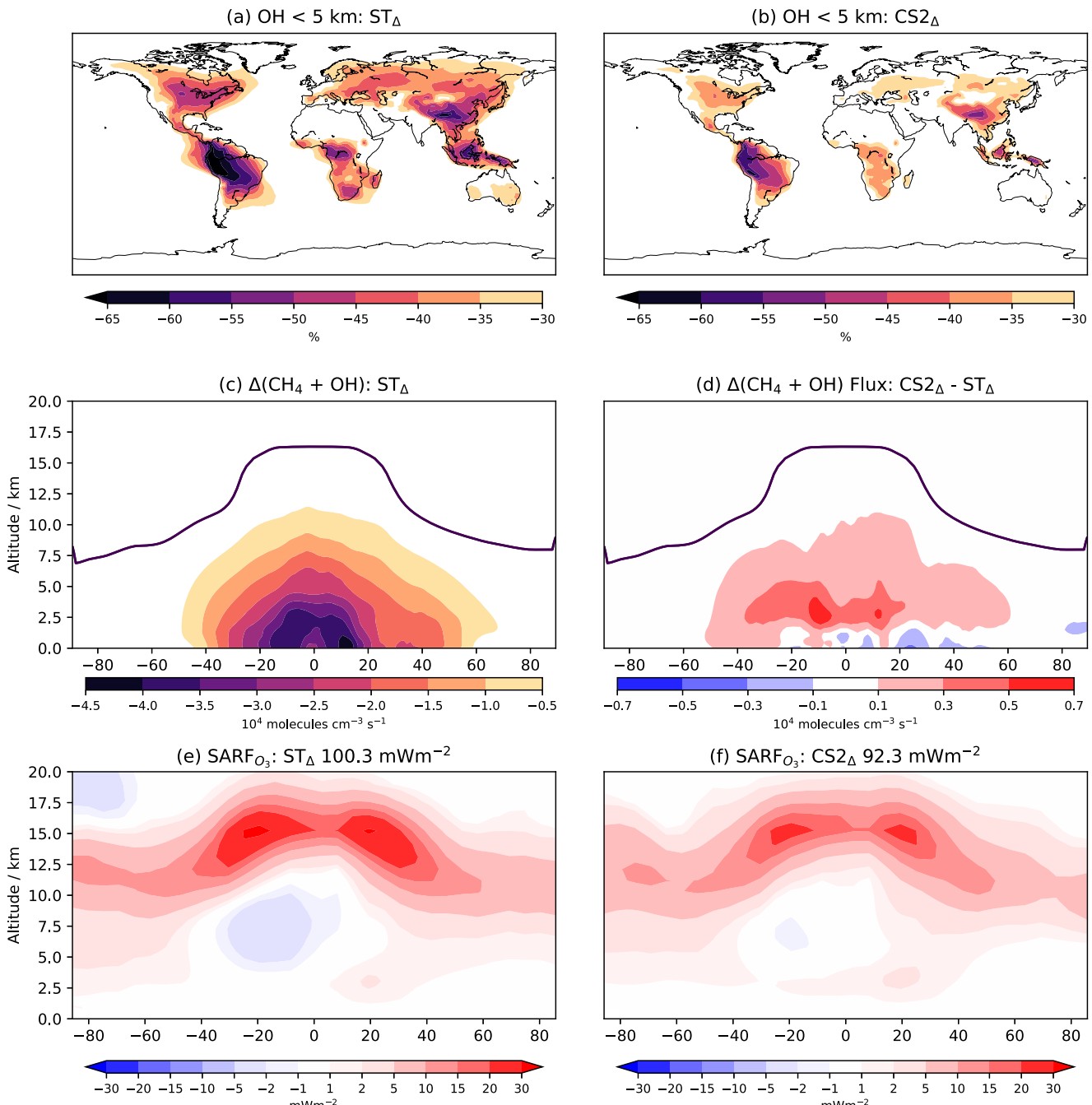

**Fig. 3 | OH, CH₄ oxidation and O₃ radiative forcing.** Percentage change in OH in lowest 5 km for (**a**) Strat-Trop (ST$_\Delta$) and (**b**) CRI-Strat 2 (CS2$_\Delta$). Zonal mean change in CH₄ oxidation flux for (**c**) ST$_\Delta$ and (**d**) CS2$_\Delta$ - ST$_\Delta$. Forcing from O₃ changes (SARF$_{O3}$) for (**e**) ST$_\Delta$ and (**f**) CS2$_\Delta$, values in title show global mean forcing.

means SOA concentrations increases by more in the FT in ST$_\Delta$ and in the PBL for CS2$_\Delta$ (Fig. 4a) while column SOA increases are greater over E$_{BVOC}$ source regions in CS2$_\Delta$ and over more remote regions, particularly the central Atlantic, in ST$_\Delta$ (Fig. 4b).

ST$_\Delta$ and CS2$_\Delta$ also differ in how the extra SOA-precursors alter the SOA size and number distribution. The greater increase of precursors within the PBL in CS2$_\Delta$ results in a larger increase in condensation to accumulation mode particles than in ST$_\Delta$. Conversely, the greater transport of precursors into the FT in ST$_\Delta$ means condensation flux to the Aitken mode increases by a greater extent. In turn this yields a greater increase in accumulation number concentration in ST$_\Delta$, via growth of Aitken particles to accumulation mode size, over much of Amazonia, central Africa and the central Atlantic.

The differences in SOA dispersion and accumulation mode number concentration between ST$_\Delta$ and CS2$_\Delta$ have direct consequences for the spatial changes in aerosol scattering and the attendant forcing. The statistically significant (95% confidence) IRF$_{DRE}$ is slightly stronger over Amazonia and central Africa in CS2$_\Delta$ but noticeably stronger over the central Atlantic in ST$_\Delta$ (Fig. 4c, d), correlating well with the difference in SOA column and aerosol number concentration. The IRF$_{DRE}$ is the single largest forcing component and the greater dispersion of additional SOA in ST$_\Delta$ leads to a 7% stronger forcing (−260 vs. −244 mWm⁻²). Similarly[30], found that following a doubling of SOA, greater transport of SOA in the EC-Earth model, compared to NorESM and ECCHAM, led to a stronger IRF$_{DRE}$.

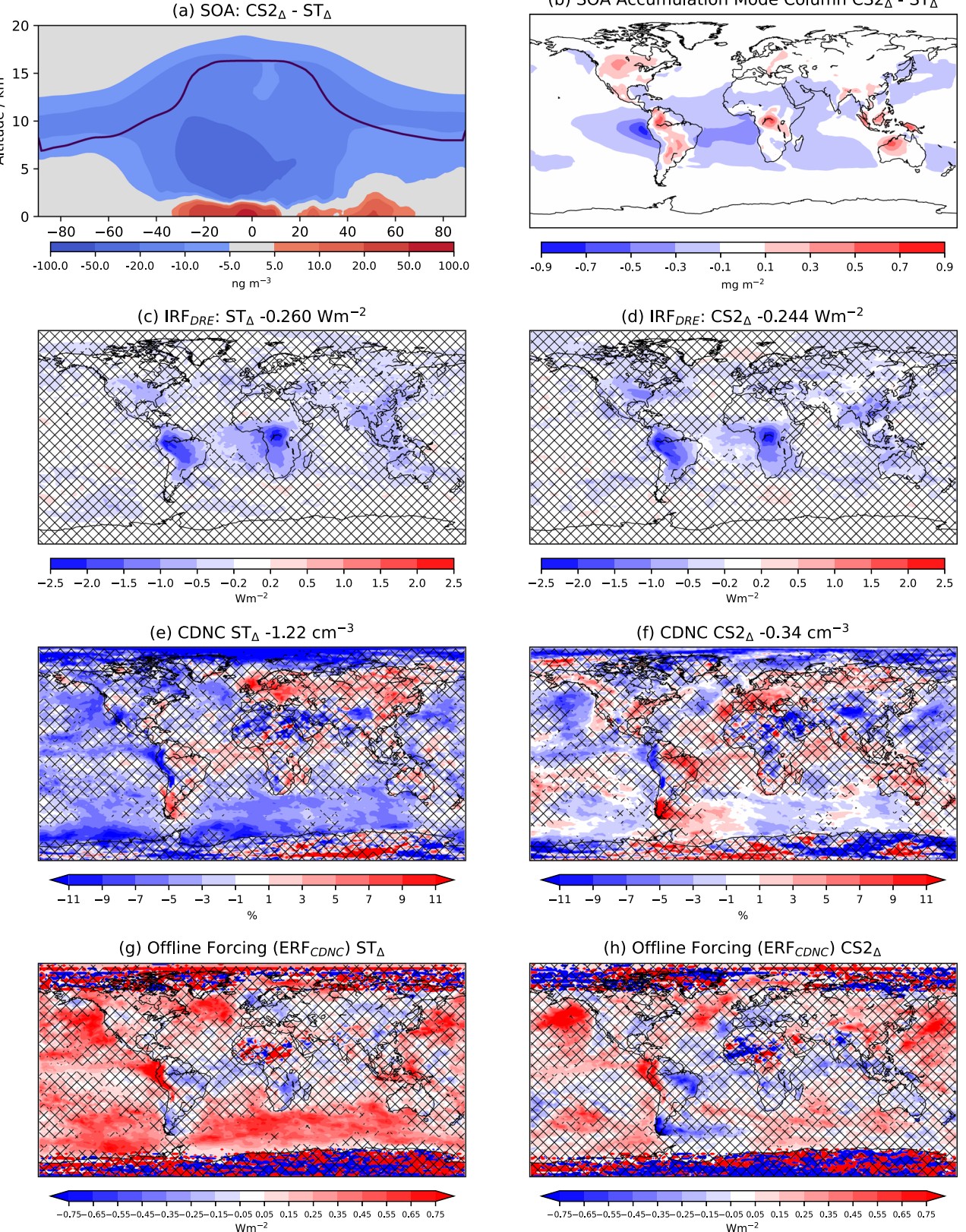

**Fig. 4 | Secondary organic aerosol (SOA) changes, aerosol scattering forcing, cloud droplet number concentration changes and associated cloud forcing.** Difference in (**a**) zonal mean SOA increase and (**b**) accumulation mode SOA column increase between CRI-Strat 2 (CS2$_\Delta$) and Strat-Trop (ST$_\Delta$). Radiative forcing from the change in the aerosol direct radiative effect (IRF$_{DRE}$) for (**c**) ST$_\Delta$ and (**d**) CS2$_\Delta$. Percentage change in vertically averaged cloud droplet number concentration (CDNC) concentration (**e**) ST$_\Delta$ and (**f**) CS2$_\Delta$ and radiative forcing due to change in CDNC (ERF$_{CDNC}$) for (**g**) ST$_\Delta$ and (**h**) CS2$_\Delta$ calculated using the offline approach of[34] (Methods). Values in titles are global mean forcing (**c**, **d**) and CDNC change (**e**, **f**). Non-hatched regions in (**c–h**) show areas where the change is statistically significant (95% confidence).

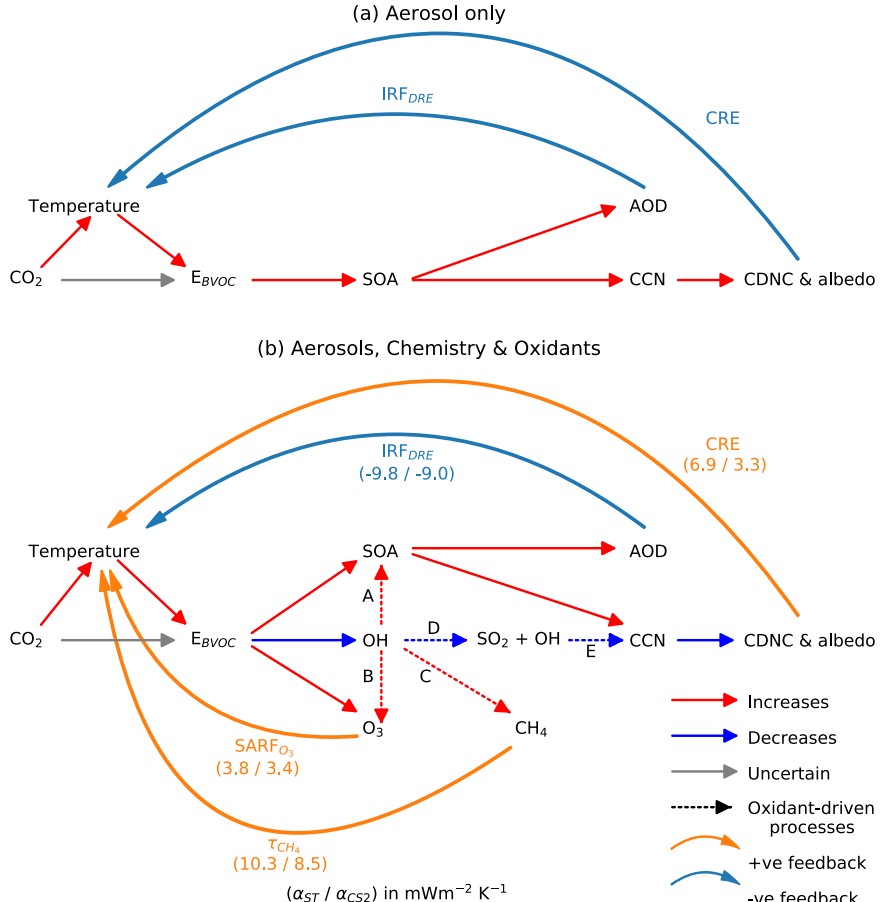

**Fig. 5 | Atmospheric composition response and BVOC feedback from an increase in CO₂ driving an increase in BVOC emissions (E$_{BVOC}$).** In (**a**) only aerosols are considered yielding a negative feedback (adapted from[23]) whereas in (**b**) chemistry and oxidants are also allowed to respond, leading to a more complex response. Aerosol optical depth (AOD) is a measure of aerosol scattering while IRF$_{DRE}$ and CRE correspond to the forcing from changes to the aerosol direct radiative effect and aerosol cloud interactions respectively. Dashed lines in (**b**) show important oxidant-driven responses including reduced OH driving (A) increased secondary organic aerosol (SOA) lifetime and climatic impact, (B) greater vertical transport of O₃ precursors and thus O₃ forcing (SARF$_{O3}$), (C) increased CH₄ lifetime ($\tau_{CH_4}$) and climatic impact and (D, E) reduction in gas phase SO₂ oxidation with attendant decreases in H₂SO₄, new particle formation, cloud condensation nuclei (CCN), cloud droplet number concentration (CDNC) and cloud albedo. The strength of the feedback for each loop is shown by the feedback factor (see Methods) for the Strat-Trop mechanism ($\alpha_{ST}$) and CRI-Strat 2 mechanism ($\alpha_{CS2}$) in parentheses.

## Cloud forcing (CRE)

In ST$_\Delta$ and CS2$_\Delta$ SOA increases drive higher cloud droplet number concentration (CDNC) over Amazonia and over the central Atlantic (Fig. 4e, f) (downwind of central Africa) following the spatial change of SOA accumulation mode aerosol, although much of the CDNC increase is not statistically significant (95% confidence).

However, statistically significant decreases in CDNC occur over large areas of the south Atlantic, south Pacific and Southern Ocean (Fig. 4e, f), regions downwind of the Amazon[31] and with high stratocumulus coverage[32]. This is driven by sulfate aerosol changes, not SOA, and the response in ST$_\Delta$ is much stronger than in CS2$_\Delta$ (Fig. 4e, f). Co-located increases in cloud droplet effective radius are also simulated and, for a given cloud liquid water content, such changes reduce cloud albedo[33]. Accordingly, both mechanisms simulate positive global SW CRE (ST$_\Delta$ 222 mWm⁻², CS2$_\Delta$ 137 mWm⁻²). Offline calculations isolating the impact of CDNC changes ([34], Methods) also find reductions in outgoing SW radiative flux (i.e., positive forcings) over the south Atlantic and Southern Ocean (Fig. 4g, h).

The CDNC decreases are driven by reduction in gas phase oxidation of SO₂ by OH to form H₂SO₄ which in turn nucleates new aerosol particles. The suppression of H₂SO₄ production is greater in ST$_\Delta$ (3.0 vs. 2.0 Tg yr⁻¹) due to the greater reduction in OH. The reduction in H₂SO₄ production and thus new particle nucleation (25 Gg yr⁻¹ ST$_\Delta$ vs. 14 Gg yr⁻¹ CS2$_\Delta$) (Fig. S3) leads to compensatory increases in aqueous phase SO₂ oxidation by H₂O₂, which only adds mass to existing particles predominantly in the accumulation mode (whereas nucleation adds to aerosol mass and number). The increase in aqueous SO₂ oxidation is reinforced by the increases in H₂O₂ from the additional BVOC loading.

The net effect is a shift in the aerosol size distribution to fewer, larger particles. Accordingly, ST$_\Delta$ exhibits a 26% decrease in Aitken mode SO₄ burden compared to 21% in CS2$_\Delta$, particularly downwind of the major biogenic emission regions (e.g., south Atlantic from Amazonia), and a more widespread decrease in Aitken mode number concentrations. Larger Aitken mode particles can activate to cloud condensation nuclei (CCN) in remote regions[30] and so their decrease reduces CDNC concentrations. The LW component of the CRE is small and very similar between the mechanisms but the net CRE of 183 mWm⁻² in ST$_\Delta$ compared to 91 mWm⁻² in CS2$_\Delta$ constitutes the largest difference between the mechanisms among the forcing components.

## The central role of oxidants

Figure 5 contrasts the feedback loops which arise when model simulations consider the impact of BVOC emissions (a) solely from aerosol changes and (b) when chemistry and oxidants change as well. The

latter yields a more complex response and highlights the central role of oxidants in influencing not only the forcing from gas phase composition changes but from aerosol and cloud property changes too.

The most noticeable difference between these paradigms is the sign of the CRE which is negative in (a) but positive in (b). This illustrates the subtle difference between the $IRF_{DRE}$ and CRE from a perturbation to $E_{BVOC}$ with interactive oxidants (as done here) and the $IRF_{DRE}$ and CRE from a perturbation to SOA or $E_{BVOC}$ with prescribed oxidants (e.g.,[22,23] respectively). In the latter case, the only way an $E_{BVOC}$ or SOA increase can impact ACI is via changes to SOA which typically results in a negative CRE (although not always, cf. EC-Earth in[30]) by providing additional condensable mass which grows aerosol particles to sizes where they can act as cloud condensation nuclei and therefore increase CDNC, with a minor contribution from enhanced aerosol nucleation. This tends to make BVOCs appear strong cooling agents. When simulating a doubling of $CO_2$, ref. [23] found the total negative aerosol forcing (direct and CRE) from the accompanying $E_{BVOC}$ increase offset 13% of the positive forcing from $CO_2$. This substantial offsetting arose from a very strong positive dependence of $E_{BVOC}$ on temperature (highest among AerChemMIP models) causing a large increase in SOA which yielded strong negative forcings, particularly from CRE, with no concomitant oxidant-driven forcing from changes to sulfate aerosol, $O_3$ and $CH_4$.

By contrast, the use of interactive oxidants here not only results in radiatively-important changes to $O_3$ and $CH_4$ but also changes in SOA transport (affecting $IRF_{DRE}$) and significant perturbations to sulfate aerosol via reduction in gas phase $SO_2$ oxidation. This reduces new particle formation and CDNC, yielding a positive CRE which outweighs the impact of increased SOA and leads to the opposite conclusion to[23].

The link between oxidants, CDNC and CRE has also been simulated in[35] where OH-suppression from increases to $CH_4$ concentration yielded CDNC reductions and a positive CRE. However, the impact of increased $H_2O_2$ (substantial from BVOC increases but less so from $CH_4$ increases) favouring aqueous phase $SO_2$ oxidation further highlights the wider range of pathways via which BVOCs can affect climate.

Fully understanding the climatic impact of BVOC emissions requires capturing as many of these oxidant-influenced interactions as possible. This is particularly important in the context of nature-based climate policies since incorrectly diagnosing their effects on climate could lead to implementation of ineffective or even counterproductive policies.

## Wider context

Changes to $CO_2$ concentration, climate (temperature, flooding, droughts) and land use policies, including well-intentioned efforts to promote biodiversity and mitigate climate change by increasing $CO_2$ sequestration (via re/afforestation or energy crops), will affect future BVOC emissions in a complex manner. Understanding how these changes will influence climate change is therefore critical for reducing uncertainty in future climate projections and ensuring that such mitigation policies are beneficial and not counterproductive.

While most prior work on the climatic impact of BVOCs has focused on the impact to aerosols and the accompanying uncertainty in BVOC-aerosol parameterisations, this work demonstrates the important coupling between aerosols, chemistry and oxidants. The necessity of using interactive (rather than prescribed) oxidants in the context of the BVOC feedback has already been demonstrated by the radiatively-important changes to $O_3$ and $CH_4$ (e.g.,[20]). By comparing the response to an $E_{BVOC}$ increase with two interactive chemical mechanisms, this study progresses beyond prior studies by identifying the wider reach of oxidants as they impact not only the forcing from gas phase composition changes but also the forcing from aerosol and cloud property changes; previously overlooked interactions. The strong dependence of the BVOC feedback on oxidants, and therefore the

chemical mechanism, demonstrates the importance of accurately representing tropospheric chemistry for determining the influence of BVOCs on climate.

Improving the understanding of the pristine PI atmosphere is important given the large degree of uncertainty in the period and the associated consequences for radiative forcing from the PI to the present day[36]. The use of the PI highlights the importance of simulated chemistry to understanding this period and its response to perturbations. It also allows this study to serve as a baseline for future work since the critical role of oxidants and sulphate aerosol identified here means the background atmospheric composition, particularly species which affect atmospheric oxidising capacity and background aerosol (e.g., $NO_x$ and $SO_x$ which are higher in the present day than the PI), will be influential in determining how changes to BVOC emissions will affect $O_3$, $CH_4$, aerosol burdens and CDNC and thus the magnitude of the opposing radiative effects which ultimately determine the climatic impact. Improvements to the description of SOA formation beyond the current fixed yield, condensation-driven approach include the adoption of more realistic processes including dimer formation from terpenes (e.g.,[37]), the reactive uptake of isoprene epoxy-diols (IEPOX)[38] and SOA formation in aqueous aerosol and cloud droplets which is believed to be comparable to gas phase SOA formation in some circumstances (e.g.,[39]). These updates may alter, to varying extents, the DRE and ACI response to a BVOC emission perturbation, thus warranting further work. The response of the DRE and ACI will be influenced by background atmospheric composition and the requirement for multiple oxidation steps for SOA-precursor formation will alter (and indeed likely accentuate) the effect of oxidants on SOA dispersion and lifetime while the complex role of $NO_x$ in IEPOX and dimer formation and the influence of aerosol composition (e.g., acidity) on IEPOX reactive uptake will drive a greater dependence on $NO_x$ and $SO_x$ and the wider background atmospheric composition.

The wide-ranging influence of oxidants and chemistry identified in this study, and the attendant dependence on atmospheric chemical composition, means a doubling of BVOC emissions in a present-day or future climate is likely to have a different climatic impact to that simulated here. Such experiments would provide further information regarding the sensitivity of BVOC's climatic impact to background atmospheric conditions and make for interesting follow up studies. When assessing the future climatic impact of a re/afforestation policy the application of the radiative efficiency or feedback factor determined using the doubling of emissions in a PI atmosphere following the CMIP6 convention may not suitable. Instead, contemporaneous background atmospheric composition must be used with the processes highlighted in this study providing a framework for such research.

Doubling BVOC emissions represents a substantial perturbation and extrapolation of this study's results to different emission scalings (e.g., 50% increase) should be performed with care since different components of the model's response are likely to scale with emissions with varying degrees of linearity. For example, the current use of a fixed SOA yield means the modelled $IRF_{DRE}$ may scale quite linearly with emissions while the non-linearity of $O_x$-$NO_x$-VOC chemistry (e.g.,[40]) means changes to OH, and thus to $CH_4$ forcing, are likely to be less linear. The complexity of the interactions and role of background atmospheric composition mean the extent of linearity can only truly be determined by further experiments.

Increasing emissions of BVOCs leads to a cascade of chemical and climatic impacts in the Earth system by driving complex changes in the distribution of oxidants with concomitant effects on the burden and lifetime of radiatively important gases, aerosols and cloud properties. Overall, we find, in a PI climate, a doubling of $E_{BVOC}$ in UKESM1 leads to increases in $O_3$ and $CH_4$ and decreases to CDNC/cloud albedo through a reduction in gas phase $SO_2$ oxidation. In ST, the combined positive forcing from these changes outweighs the negative forcing arising

from the scattering of radiation from enhanced SOA, yielding a positive feedback. However, when a state-of-the-science chemistry scheme (CS2), featuring recent developments in isoprene chemistry, is used the net positive feedback is 43% smaller. The central driver of this difference is a smaller reduction in oxidants and attendant smaller increases in $CH_4$ and smaller decreases in gas phase $SO_2$ oxidation, CDNC and cloud albedo. The smaller oxidant depletion also limits the transport of $O_3$-precursors up to the upper-troposphere, where $O_3$ is most potent as a GHG, yielding a smaller positive forcing despite a greater increase in tropospheric $O_3$ burden. The wide-scale transport of SOA from the enhanced $E_{BVOC}$ is lower in CS2 following the lower oxidant depletion, yielding a smaller negative aerosol and cloud forcing, but this effect is outweighed by the diminished positive forcings from $CH_4$, cloud albedo and $O_3$. Thus, we demonstrate the important coupling between aerosols, chemistry, and oxidants in determining the climatic impact of BVOC emissions.

## Methods

### Model runs

All model runs were performed for 45 years (15 years spin up, 30 years analysis) with pre-industrial timeslice conditions using the UKESM1-AMIP setup at a horizontal resolution of 1.25° × 1.875° with 85 vertical levels up to 85 km[41]. All simulations had fully interactive stratospheric and tropospheric chemistry, including interactive oxidants, using either the Strat-Trop (ST) mechanism[28] or the CRI-STRAT 2 (CS2) mechanism[29]. The simulations used the GLOMAP-mode aerosol scheme which simulates sulfate ($SO_4$), sea-salt (SS), black carbon (BC), primary organic aerosol (POA), secondary organic aerosol (SOA) and dust but not nitrate aerosol[42,43]. In this setup, the model tracks the mass concentration of each mode present in each component (e.g. $SO_4$ nucleation mode) and the total particle number concentration for the nucleation, Aitken (soluble and insoluble), accumulation and coarse modes.

Emissions of well-mixed greenhouse gases (WMGHGs), such as methane ($CH_4$) and $CO_2$, were not simulated; rather, prescribed lower boundary conditions at PI levels were applied for $CO_2$ (284 ppm), $CH_4$ (808 ppb) and $N_2O$ (273 ppb), consistent with control runs of UKESM1's contributions to AerChemMIP[44].

The setup of these runs followed the AerChemMIP protocol[44] to allow calculation of the ERF. Fields for SSTs, SI, ocean biogeochemistry (DMS and chlorophyll) and land cover were taken from monthly mean climatologies derived from 30 years of output of the UKESM1 fully-coupled pre-industrial control experiment (piControl) discussed in[45]. Timeslice PI anthropogenic and biomass burning emissions were taken from the CEDS dataset[46,47] respectively. While the atmosphere-only setup with fixed SSTs does constrain the wider Earth system response (for example aerosol-driven changes to PAR cannot change land cover via fertilisation of additional vegetation), it does reduce the noise which would occur with a coupled ocean. Importantly it also allows this study's results to be directly comparable to other studies such as the emission perturbation runs in AerChemMIP. The use of ERF, as opposed to other definitions of radiative forcing such as instantaneous radiative forcing, allows the inclusion of stratospheric temperature adjustments but also rapid adjustments in the troposphere including temperature, water vapour, clouds, and land surface temperature[35].

All terrestrial biogenic emissions, except isoprene and MT, were based on 2001–2010 climatologies from Model of Emissions of Gases and Aerosols from Nature under the Monitoring Atmospheric Composition and Climate project (MEGAN-MACC) version 2.1[48]. Oceanic emissions were from the POET 1990 dataset[49]. Oceanic DMS emissions were calculated from seawater DMS concentrations[45] which were prescribed from the fully coupled UKESM1 PI control run.

As in the UKESM1 runs for AerChemMIP, isoprene and MT emissions were calculated using the iBVOC emissions system[50] which calculates the emissions interactively based on temperature, $CO_2$, plant functional type and photosynthetic activity. The use of iBVOC allows for a more faithful estimate of pre-industrial emissions of biogenic species compared to using present-day emissions inventories such as MEGAN-MACC[9] since iBVOC considers the PI land use and atmospheric conditions such as lower $CO_2$. In the CS2 runs, the MT emissions calculated by the iBVOC system were split into α-pinene and β-pinene in a 2:1 ratio as in previous studies using the CRI mechanisms[29,51].

### Chemical mechanisms

The scale of tropospheric chemistry (~19,000 reactions for organic species alone in the near-explicit Master Chemical Mechanism (MCM)[52]; prevents explicit simulation and necessitates the use of condensed mechanisms which reduce complexity by lumping chemical species together and considering only the most important reactions.

The Strat-Trop (ST) and CRI-Strat 2 (CS2) chemical mechanisms are described in detail in[28,29] respectively with a full description of every tropospheric chemistry reaction in CS2 also available at http://cri.york.ac.uk/home.htt (last accessed 5th June 2022). ST considers 73 species and 305 reactions while CS2 has 228 species and 766 reactions with the bulk of the added complexity coming from a wider range of organic species (Tables 2 and S01[51]). ST does not feature the CS2 species $C_2H_2$, $C_2H_4$, $C_3H_6$, $C_2H_5OH$, $C_2H_5CHO$ and methyl ethyl ketone but does add their emissions to species it does consider (e.g., emissions of $C_2H_4$ are included in $C_2H_6$ in ST). Some species are omitted entirely by ST and are only included in CS2. These are butane, butene, benzene, toluene, oxylene, formic acid and ethanoic acid (Table 3[51]).

CS2 is based on the tropospheric chemistry scheme CRI v2.2[53] which is traceable to the latest version of the MCM (v3.3.1) and conserves its ozone forming potential.

As illustrated in Fig. 1, a major difference between the mechanisms is the inclusion of the H-shift pathways of ISOPOO ($C_5H_9O_3$). ST features isoprene chemistry from[54] where ISOPOO forms the isoprene hydroperoxide (ISOPOOH) via reaction with $HO_2$ and methacrolein the major product from reaction of ISOPOO with NO, $NO_3$ and other peroxy radicals ($RO_2$). By contrast, CS2 also features the 1,4- and 1,6-H-shift reactions of ISOPOO. CS2 also simulates organonitrate formation from a wide range of $RO_2$ whereas ST uses the methyl nitrate ($CH_3ONO_2$), and isoprene nitrate ($C_5H_9NO_3$) and nitrooxy aldehyde ($C_2H_3NO_4$) to represent all organonitrates. CS2 simulates 50–100% higher OH concentration in terrestrial tropical regions than ST, improving model performance for OH, isoprene and monoterpenes[29,53]. CS2 is comparable to other more advanced chemical mechanisms such as the CalTech reduced isoprene scheme[55] but the effect of this chemistry on the climatic impact of BVOCs has not been assessed. Isoprene oxidation also produces the chemically-inert species, $Sec\_Org_{ISOP}$ (Fig. 1), which condenses onto aerosol.

For monoterpenes ST features a single tracer (MT) whose oxidation by $O_3$, OH and $NO_3$ produces only a chemically-inert species, $Sec\_Org_{MT}$ (Fig. 1), which condenses onto aerosol or nucleates new aerosol with sulfuric acid. This lack of further chemistry means MT only acts as an oxidant sink rather than behaving as reactive organic carbon (ROC)[56]. In CS2, monoterpene chemistry features oxidation of α-pinene and β-pinene (a sink of oxidants) which produces both $Sec\_Org_{MT}$ and other chemically active products. These oxidation products undergo further chemical reactions[57] (Fig. 1). The transport of these oxidation products can lead to the regeneration of $O_3$ and OH away from emission sources, offsetting some of the oxidant depletion from initial oxidation of the monoterpenes, with associated effects on $CH_4$ and aerosol.

In the PI, CS2 simulates an extra ~5 TgC yr$^{-1}$ of ROC emissions than ST due to the wider range of emitted VOCs considered by CS2[51].

In addition, CS2 features an extra ~120 TgC yr$^{-1}$ of reactive organic carbon produced in the atmosphere in the form of 1st generation oxidation products from monoterpenes since monoterpene oxidation in ST does not produce any chemically-active species (Fig. 1). Prior mechanistic analysis has identified this additional ROC to lead to lower surface OH but greater OH in the tropical lower free troposphere (FT)[51].

UKESM performance using ST and CS2 was evaluated against present day observational data of BVOCs and other important chemical species from surface sites, flight campaigns and satellites with a full description in[29]. Relative to ST, CS2 reduced the model's high isoprene and monoterpene bias at the surface by increasing the local OH concentration. CS2 also yielded substantial improvements in isoprene column over Amazonia, Africa and southeast Asia.

The rate constant for the reaction of MT + NO$_3$ in ST was corrected from the erroneously high expression of $1.19 \times 10^{-12}e^{925/T}$ to $1.19 \times 10^{-12}e^{490/T}$, bringing it into line with the IUPAC preferred value (https://iupac-aeris.ipsl.fr/htdocs/datasheets/pdf/NO3_VOC9_NO3_apinene.pdf, last accessed 14th September 2021) for α-pinene on which the ST tracer MT is based. This results in a reduction in the rate constant of ~80%, but as NO$_3$ is a minor sink for monoterpenes, this change does not have a huge impact on aerosol formation.

## SOA scheme improvements

The UKESM1 contributions to AerChemMIP (which also used the Strat-Trop chemical mechanism) simulated SOA production only from monoterpene oxidation with a doubled molar yield of 26% (28.6% mass yield) to account for the lack of SOA production principally from isoprene but also other VOCs[43]. However, as a greater fraction of monoterpenes are produced in high latitude forests compared to isoprene[9], this approach skewed SOA production to higher latitudes with implications for SOA lifetime and climatic impact. Nucleation of new particles from the clustering of oxidised organic species and sulfuric acid was also omitted in the UKESM1 simulations for AerChemMIP.

In this current study, the description of SOA formation was improved from that used by UKESM in AerChemMIP to include SOA production from isoprene as well as monoterpenes and aerosol nucleation in the boundary layer from Sec_Org$_{MT}$ and H$_2$SO$_4$. Inert SOA-precursors were produced from monoterpenes (Sec_Org$_{MT}$) at the original molar yield of 13% (14.3% mass yield) (Eq. (1)) and isoprene (Sec_Org$_{Isop}$ 3% molar yield; 3.3% mass yield) (Eq. (2)). SOA-precursors from both monoterpenes and isoprene could condense onto existing aerosol while nucleation of new particles via the clustering of H$_2$SO$_4$ and Sec_Org$_{MT}$ was also simulated following the scheme of[58] (Eq. (3)) but constrained to the model boundary layer. The inclusion of isoprene SOA and boundary layer nucleation (BLN) represent improvements over the standard UKESM1 model setup used for AerChemMIP (e.g.,[20]). The formation of SOA was the same in ST and CS2.

$$MT + O_x \rightarrow 0.13\,Sec\_Org_{MT} \rightarrow Condensation\ or\ Nucleation \quad (1)$$

$$Isoprene + O_x \rightarrow 0.03\,Sec\_Org_{ISOP} \rightarrow Condensation \quad (2)$$

$$J = k[H_2SO_4][Sec\_Org_{MT}]\ where\ k = 5 \times 10^{-13}\ molecules\ cm^{-3}s^{-1} \quad (3)$$

$$(O_x = OH, O_3, NO_3)$$

The change in SOA precursor yields leads to total organic aerosol (primary + secondary) burdens which are 9% and 17% higher in the ST$_{con}$ and ST$_{2x}$ simulations in this study compared to the corresponding PI control and 2xE$_{BVOC}$ UKESM1 simulations in AerChemMIP.

## Forcing definitions

For each mechanism pair, the ERF is defined as the difference in TOA net radiative flux (Eq. (4))

$$ERF = \Delta N = N_{2x} - N_{con} \quad (4)$$

Following the approach of[35,59], the ERF can be decomposed in aerosol direct radiative effects (IRF$_{DRE}$) (Eq. (5)), aerosol-cloud effects (CRE) (Eq. (6)), and clear-sky effects (CS) (Eq. (7)).

$$IRF_{DRE} = \Delta(N - N_{clean}) \quad (5)$$

$$\Delta CRE = \Delta(N_{clean} - N_{clear,clean}) \quad (6)$$

$$ERF_{CS} = \Delta(N_{clear,clean}) \quad (7)$$

N$_{clean}$ is the net flux excluding scattering and absorption by aerosols, and, N$_{clear,clean}$ is the flux excluding scattering and absorption by aerosols and clouds. Thus, the IRF$_{DRE}$ corresponds to the difference in net TOA radiative flux due solely to the scattering and absorption of aerosols (changes to land surface albedo are negligible due to prescribed land use) while the CRE reflects changes to cloud forcing via aerosol indirect effects. The clear sky forcing corresponds to change due to the absorption and emission of radiation by gas phase species.

The prescribed surface concentration of CH$_4$ in the model setup significantly constrains the response of CH$_4$ concentrations to oxidant perturbations and thus the radiative effect. However, the change in CH$_4$ concentration which would have occurred had surface CH$_4$ concentration not been constrained can be diagnosed (Eq. (8)).

$$\frac{\Delta C}{C} = \left(\frac{\Delta \tau}{\tau} + 1\right)^f - 1 \quad (8)$$

Where C is the CH$_4$ concentration, τ is the methane lifetime and f is the feedback of methane on its own lifetime[60] taken as 1.28 for the pre-industrial period[35]. The forcing due to the change in CH$_4$ concentration was then calculated using the approach in[61] using the baseline concentrations of CH$_4$ and N$_2$O of 808 ppb and 273 ppb respectively. Following[20], this forcing was then scaled by 1.52 to account for the additional chemical production of ozone and stratospheric water vapour.

Unlike methane, O$_3$ concentrations can respond to changes in E$_{BVOC}$, and the resulting forcing is included in the clear sky forcing component, ERF$_{CS}$. The forcing from ozone changes was isolated using the radiative kernel from[62] as in[20] which yielded the stratospheric-temperature adjusted radiative forcing (SARF$_{O3}$).

## Offline CDNC forcing calculation

Offline radiative flux calculations were performed to calculate the forcing due to changes in CDNC alone (ERF$_{CDNC}$). Monthly mean values were used for all variables for these calculations. This followed the technique described in[34,63] for TOA fluxes and used CDNC, total cloud fraction (f$_c$, calculated using maximum random overlap), in-cloud (as opposed to all-sky) liquid water path (LWP$_{ic}$), SW clear-sky upwelling flux at TOA (F$_{SW}^{clear-sky}$), SW downwelling flux at TOA (F$_{sw,down}$) and the surface albedo (A$_{surf}$) as inputs. The approach used here differs slightly to those studies due to the inclusion here of F$_{SW}^{clear-sky}$ from the model for the clear-sky regions rather than assuming a constant transmissivity. A transmissivity of 0.89 was use above cloud. Multiple scattering between the surface and cloud was also included here following[64]. A$_{surf}$ was calculated by dividing the upwelling clear-sky SW surface fluxes by the corresponding downwelling fluxes. LWP$_{in-cloud}$ is the LWP from the cloudy regions only

and was calculated by dividing the all-sky LWP data (as output by the model) by $f_c$ (e.g., as in[65]).

$ERF_{CDNC}$ was calculated firstly by using the control (con) values as a baseline for the SW TOA flux ($F_{SW}$) calculation and then calculating the difference between this and an $F_{sw}$ value calculated using the 2× BVOC (2×) values for CDNC and control values for everything else (Eq. (9)).

$$ERF_{CDNC,con\ base} = F_{SW}\left(CDNC_{2x}, f_{c,con}, LWP_{ic,con}, F^{clear-sky}_{SW,con}, F_{SW,down,con}, A_{surf,con}\right)$$
$$- F_{SW}\left(CDNC_{con}, f_{c,con}, LWP_{ic,con}, F^{clear-sky}_{SW,con}, F_{SW,down,con}, A_{surf,con}\right) \tag{9}$$

Then the 2xBVOC run was used as a baseline and the CDNC from the control substituted in Eq. (10).

$$ERF_{CDNC,2x\ base} = F_{SW}\left(CDNC_{2x}, f_{c,2x}, LWP_{ic,2x}, F^{clear-sky}_{SW,2x}, F_{SW,down,2x}, A_{surf,2x}\right)$$
$$- F_{SW}\left(CDNC_{con}, f_{c,2x}, LWP_{ic,2x}, F^{clear-sky}_{SW,2x}, F_{SW,down,2x}, A_{surf,2x}\right) \tag{10}$$

An overall value for $ERF_{CDNC}$ was calculated as the average of $ERF_{CDNC,con\ base}$ and $ERF_{CDNC,\ 2x\ base}$.

### Feedback factor

For a given forcing ΔF, the resultant change to TOA radiative imbalance, ΔN, can be expressed by $\Delta N = \Delta F + \alpha \Delta T$ where α is the climate feedback parameter and represents the rate of change of the TOA radiative imbalance with respect to the global mean change in surface temperature, ΔT. α can be decomposed into individual feedback terms, $\alpha_i$, arising from changes to different climate variables, $C_i$ (Eq. (11)).

$$\alpha = \frac{d\Delta N}{d\Delta T} = \sum_i \frac{\partial \Delta N}{\partial \Delta C_i}\frac{\partial \Delta C_i}{\partial \Delta T} = \sum_i \alpha_i \tag{11}$$

In this study the climate variable of interest is $E_{BVOC}$. The corresponding feedback factor, $\alpha_{BVOC}$, can be considered as the forcing arising from the change in $E_{BVOC}$ in response to a temperature change (Eq. 12).

$$\alpha_{BVOC} = \frac{\partial \Delta N}{\partial \Delta E_{BVOC}}\frac{\partial \Delta E_{BVOC}}{\partial \Delta T} = \phi_{BVOC}\gamma_{BVOC} \tag{12}$$

Where $\phi_{BVOC}$ is the radiative efficiency per unit change in emissions (i.e., the change in TOA radiative imbalance per unit change in emissions with typical units of $Wm^{-2}$ $(Tg\ yr^{-1})^{-1}$) and $\gamma_{BVOC}$ is the change in $E_{BVOC}$ with climate ($Tg\ yr^{-1}\ K^{-1}$).

$\phi_{BVOC}$ is calculated by dividing the radiative forcing diagnosed from the timeslice model simulation pairs ($ST_{con}$ & $ST_{2x}$, $CS2_{con}$ & $CS2_{2x}$) by the change in emissions.

$\gamma_{BVOC}$ is diagnosed from a pair of timeslice model simulations: the piControl which simulates an 1850s atmosphere and the abrupt-4xCO$_2$ which is initialised from the piControl before atmospheric $CO_2$ concentrations are instantly quadrupled. These simulations were run for 150 years as part of the AerChemMIP project[44] and the changes in temperature and emissions were calculated from the mean of years 121–150. The change in $E_{BVOC}$ per unit temperature change was then calculated.

### Nomenclature

The terms used to represent the response of an atmospheric parameter for a given mechanism are defined in Eqs. (13) and (14) while

Eq. (15) shows the difference between two responses.

$$ST_\Delta = ST_{2x} - ST_{con} \tag{13}$$

$$CS2_\Delta = CS2_{2x} - CS2_{con} \tag{14}$$

$$CS2_\Delta - ST_\Delta = (CS2_{2x} - CS2_{con}) - (ST_{2x} - ST_{con}) \tag{15}$$

## Data availability

The UKESM1 data generated in this study have been deposited in the University of Cambridge Apollo database https://doi.org/10.17863/CAM.83526. The data are available for all users.

## Code availability

Due to intellectual property right restrictions, we cannot provide either the source code or documentation papers for the UM. The Met Office United Model is available for use under licence. A number of research organisations and national meteorological services use the UM in collaboration with the UK Met Office to undertake atmospheric process research, produce forecasts, develop the UM code, and build and evaluate Earth system models. For further information on how to apply for a licence, see https://www.metoffice.gov.uk/research/approach/modelling-systems/unified-model (last access: 18 July 2022).

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

## Acknowledgements

Vice-Chancellor's Award from the Cambridge Trust (J.W.). NERC through the University of Cambridge ESS-DT (Y.M.S.). NERC PROMOTE (grant no. NE/P016383/1) (S.A.N., A.T.A.). Support from NERC and NCAS through the ACSIS project (N.L.A. and A.T.A.). National Environmental Research Council (NERC) national capability grant for The North Atlantic Climate System Integrated Study (ACSIS) program (grant NE/N018001/1) via NCAS, and by the ADVANCE (Aerosol-cloud-climate interactions derived from Degassing VolcANiC Eruptions; NE/T006897/1) program (D.P.G.). This work used Monsoon2, a collaborative high-performance computing facility funded by the Met Office and the Natural Environment Research Council. This work used JASMIN, the UK collaborative data analysis facility.

## Author contributions

J.W. set up and executed the simulations with help from N.L.A., S.A.N., Y.M.S. and C.E.S. J.W. analysed the output with input from A.T.A., S.A.N., Y.M.S., P.T.G. and D.P.G. D.P.G. performed the offline CDNC forcing calculations.

## Competing interests

The authors declare no competing interests.
