## [Peer Review File · Nature Communications]

REVIEWER COMMENTS

Reviewer #1 (Remarks to the Author):

Weber et al. present a modeling study of BVOC chemistry in a climate context. The study is well written. But I note a few major comments below that suggest one needs to be cautious about complex chemistry climate feedbacks without observations.

1. The study compares two different atmospheric chemistry mechanisms using a pure modeling effort. But there are no comparisons to any observations. While I agree PI simulations cannot be compared directly with observations, one can look at field campaigns like the Amazon during wet season that approaches PI conditions. Both GoAmazon2014/5 from DOE G-1 aircraft and ACRIDICON-CHUVA with German HALO aircraft probed atmospheric chemistry over the Amazon.

<https://acp.copernicus.org/articles/18/6461/2018/>

2. While the results of this study are based on preindustrial simulations with a doubling of EBVOC, the conclusions and implications focus on the future. It is important to conduct simulations in the present day and future scenarios as well where BOVC-chemistry-climate interactions are very different than preindustrial conditions due to higher pollution and NO_x levels. In my opinion, preindustrial simulations should not be used to derive conclusions and implications for the future.

Minor comment:

2. OH recycling is shown to be important in the CS₂ mechanism affecting the conclusions of this study. But over the Amazon, soil NO_x emissions even (20-50 ppt) could efficiently recycle OH near the surface without the need for OH recycling mechanisms. For example, see the following modeling paper: <https://www.nature.com/articles/s41467-019-08909-4>

AT the very least the role of soil NO_x in OH recycling could be acknowledged.

Reviewer #2 (Remarks to the Author):

This paper is clearly written and illustrates an important source of uncertainty in climate modeling that is not generally represented. This work shows how differences in the complexity of chemical mechanisms in an Earth System Model can lead to different climate responses. These differences are due to differences in the oxidation of biogenic compounds (isoprene and terpenes) and resulting gas-phase chemistry, as well as the resulting formation of secondary organic aerosols. The impact of the chemistry of biogenic compounds is quantified by performing simulations with doubled biogenic emissions for both (simpler and complex) chemistry schemes in the UK Earth System Model. The paper describes clearly the

important drivers affecting the chemistry effects and climate forcings and feedbacks. Climate models frequently ignore the impacts that atmospheric chemistry has on climate due to the computational cost of including them, therefore this paper is extremely valuable for highlighting the importance of adequately representing the oxidation of some of the most abundant volatile organic compounds in the atmosphere.

I feel the paper could be improved by addressing the following points:

Chemical Mechanisms - it would be helpful to have more information in the main text about the differences between ST and CS₂, as these are the basis of most of your figures and results. A table listing the number of species and reactions for different classes of compounds (small HCs, isoprene, terpenes, SOA formation, etc.) for each mechanism would help illustrate the differences.

Figure 4 illustrates the feedbacks determined from a model with specified oxidants (a) and with detailed chemistry (b). I think it is a valuable figure and an excellent way to illustrate the complexity of the true atmosphere. However, it seems a bit incongruous as the paper does not use any simulations with specified oxidants (aerosols only). The work presented compares simpler and more complex chemistry, so I would expect such a diagram to illustrate the work done for this paper. Perhaps including a 3rd diagram in the figure showing the ST mechanism, assuming the current panel b represents the CS₂ mechanism. This would also address my previous point about showing how the ST and CS₂ mechanisms differ.

A paper that does show the differences in organic aerosols between using specified oxidants and detailed is Tilmes et al., JAMES, 2019. While this is not the primary point of your paper, it might be worth referencing.

Tilmes, S., Hodzic, A., Emmons, L. K., Mills, M. J., Gettelman, A., Kinnison, D. E., et al. (2019). Climate forcing and trends of organic aerosols in the Community Earth System Model (CESM2). *Journal of Advances in Modeling Earth Systems*, 11. <https://doi.org/10.1029/2019MS001827>

Minor points:

I.117 - give chemical formula for ISOPOO

I.151 - 'advanced' in greek letters!

I.178 - define SARF - RF is defined, but what does SA stand for?

Fig.1 caption: need to explain x-axis labels and which y-axis they correspond to.

Data availability: doi link is invalid (or not yet working).

Reviewer #3 (Remarks to the Author):

This manuscript describes key differences between two sets of paired of climate simulations designed to test the effects of the models' oxidation mechanisms for biogenic volatile organic compounds (BVOCs) on their simulated climate feedbacks. Huge advances have recently been made in the complexity and accuracy of BVOC oxidation mechanisms, but these are often only incorporated into atmospheric chemistry models; climate and earth-system models still tend to use simple and potentially outdated chemistry schemes. The analysis in this work is therefore a highly important diagnosis of the difference that these new BVOC oxidation mechanisms can make to climate simulations. Furthermore, while most similar studies focus on the effects of BVOC oxidation on secondary organic aerosol (SOA) and ozone formation, the authors of this study notably find that differences in tropospheric oxidation capacity between BVOC mechanisms can have stronger effects on climate, through changes in the methane lifetime and SO₂ oxidation.

The methodology in the manuscript is clear and the simulations and their workup appear well executed. My main complaints (detailed below) are about the remaining simplifications in the more complex chemistry scheme employed here, particularly as they relate to SOA formation. Much about BVOC oxidation and SOA formation remains uncertain and/or is understandably too complicated to incorporate in detail into a model of this scale, and I am not trying to advocate that the authors re-run their simulations with more complex chemistry added, but more discussion and emphasis is needed on the potential for the simplifications in this model to obscure or alter the important feedbacks of BVOC on aerosol formation and thus on climate. Similarly, it would be useful to discuss in more detail the ways in which the particular setup of these simulations -- especially the use of the pre-industrial baseline and the use of a doubling in BVOC emissions to diagnose feedbacks -- might differ from other relevant changes (e.g., marginal changes in BVOC emissions, or present-day conditions), to avoid readers extrapolating these results to conditions in which they might not be strictly applicable.

Specific questions:

L 128: Why are preindustrial conditions chosen as a baseline, and to what extent can these results be extrapolated to changes in present-day BVOC emissions? The effects of BVOC emissions are likely very different in the present-day atmosphere -- in which anthropogenic NO_x increases the ozone production efficiency of BVOCs and anthropogenic SO_x increases the SOA production efficiency of BVOCs -- than they were in the preindustrial atmosphere, and in ways that are strongly affected by the chemical mechanism used (e.g. the treatment of organonitrates, which play a much larger role in the higher-NO_x present-day atmosphere than they did in the preindustrial atmosphere).

Similarly, why is a doubling of BVOC emissions chosen as the perturbation, rather than, say, a marginal change or a zeroing of BVOC emissions? It seems that a zeroing would provide a more precise description of the actual impact of BVOC chemistry, while a marginal change would provide a more realistic baseline for extrapolation to actual likely changes. Because the subsequent chemical outcomes of these perturbations can so often be non-linear, the effects of a global BVOC doubling may differ substantially from marginal effects, or from global reductions in BVOC emissions. (I am not trying to say that the authors need to go back to square one and run new simulations, but rather that some attention should be paid in the manuscript to the consequences of these particular choices and the extent to which these results can be extrapolated).

L 151: "advanced" should be in Latin not Greek alphabet :)

L 162-163: What are these 180 TgC of extra emitted reactive compounds? Do directly emitted non-(isoprene+MT) compounds contribute to the changes between ST and CS2, and if so, how much and what compounds? Also, it is stated here that the higher VOC emissions in CS2 leads to greater OH, while below (L 197) it is stated that an increase in VOC emissions depletes OH. Is the greater OH in CS2 really attributable to the 180 TgC of extra emitted VOCs, or to mechanistic differences?

L 203: "greater increase OH production" should probably have an "in" in there?

L 204: Why does CS2 have higher OH production from hydroperoxide (ROOH) photolysis than ST? The reasoning for the increase OH production from HO₂+NO is well explained in L 206-210, but the ROOH contribution isn't described.

L 234-242: It is surprising that PAN-driven changes in NO_x transport between the two mechanisms dominates over the effects of NO_x depletion from organonitrates. To what extent do the mechanisms' treatments of organonitrates differ, and to what extent do both the organonitrates and their differences between the mechanisms contribute to BVOC-driven changes in ozone formation? If the CS2 mechanism included rapid aqueous hydrolysis of tertiary nitrates (Vasquez et al., 2020; DOI: 10.1073/pnas.2017442117), would the conclusions change?

L 330: "Typically" appears twice.

L 492-496: While SOA formation schemes in complex models such as these must necessarily be simplified, the simplifications here, employed even in the updated CS2 mechanism, seem potentially problematic in ways that could influence the paper's conclusions:

(1) SOA precursors are not produced in fixed yields, but rather depend on the fates of peroxy radicals in the BVOC oxidation cascade. For example, the dominant SOA precursor from isoprene is presumed to be IEPOX, formed in the RO₂ + HO₂ pathway, while dimerization

from RO2 + RO2 reactions is thought to be the most efficient SOA-forming pathway for terpenes. Thus, the yields of SOA precursors should be allowed to vary with the BVOC oxidation pathways; using a fixed yield instead shuts off this potential avenue for changes in SOA yields. (Furthermore, SOA precursors are not inert, as represented here, but themselves have oxidation pathways that may limit their contribution to aerosol in regions with higher oxidative capacity or lower preexisting aerosol loadings).

(2) SOA production from potential precursors is often not a given, and not condensation driven, but rather dependent upon other aerosol conditions, especially for reactive uptake. For example, IEPOX is not guaranteed to produce SOA, and its SOA formation is reactive, not condensation-driven. This reactivity depends on aerosol liquid water content, acidity, sulfate content (both anthropogenically driven) and organic shell (see, e.g., Nah et al. 2019, DOI: 10.1016/j.atmosenv.2019.116953 and Gaston et al. 2014, DOI: 10.1021/es5034266). Each of these dependencies can provide self-limiting feedbacks both from the extent of SOA formation and from other features of BVOC oxidation (e.g. the changes in SO2 fate highlighted in this manuscript).

(3) Relatedly, the SOA yields used here may not apply to preindustrial conditions; for example, a 3% yield from isoprene was found to fit observations in the present-day Southeast US well, but this is a region with relatively acidic, sulfate-rich aerosol that would probably lead to higher IEPOX uptake than in most preindustrial environments (Marais et al. 2016, DOI: 10.5194/acp-16-1603-2016).

(4) A key part of the conclusions in this manuscript depends on the timescale by which SOA is produced following BVOC emission, because lower OH in ST leads to slower BVOC oxidation and a greater permeation of SOA precursors to the free troposphere (L 251-255). However, this requires that the timescale of SOA formation be correctly calibrated to begin with. It isn't clear from the condensation reactions listed here what timescale is employed in these simulations, but one certainty is that IEPOX formation, and thus SOA formation from isoprene, is not a one-step oxidation process as suggested here; instead, it requires two generations of OH oxidation, which may make a substantial difference to the fraction of IEPOX that reaches the free troposphere relative to the inert precursor represented here.

(5) Whether or not SOA formation is condensation-driven, it is not an irreversible process, as it seems to be in these simulations; dilution of air masses in which SOA formed can cause the revolatilization of SOA driven by condensation equilibria, and condensed-phase or heterogeneous reactions of aerosols can cause fragmentation of organic SOA constituents.

I know this is a long list of complaints about the SOA schemes here -- again, I'm not trying to say the authors should change and redo their entire simulations, but much more nuanced discussion is required of the uncertainties and potential for additional factors to influence BVOC SOA formation and its dependencies.

Response to Reviewers

We are grateful to the reviewers for their comments and efforts which have helped us improve this manuscript. We have responded to each reviewers' comments sequentially below with italicised text showing the reviewer's comments and plain text showing our response. Text which has been added to the manuscript is coloured red. Original manuscript text is in blue and any text which has been removed from the manuscript is blue and has been struck through. The locations of changes are stated. Where reviewer comments were unnumbered, we have added numbers for ease of reference, e.g. (1). We hope these revisions address the concerns of the reviewers.

Where literature has been cited in our response, we have given the full reference at the document.

Reviewer #1 (Remarks to the Author):

Weber et al. present a modeling study of BVOC chemistry in a climate context. The study is well written. But I note a few major comments below that suggest one needs to be cautious about complex chemistry climate feedbacks without observations.

1. The study compares two different atmospheric chemistry mechanisms using a pure modeling effort. But there are no comparisons to any observations. While I agree PI simulations cannot be compared directly with observations, one can look at field campaigns like the Amazon during wet season that approaches PI conditions. Both GoAmazon2014/5 from DOE G-1 aircraft and ACRIDICON-CHUVA with German HALO aircraft probed atmospheric chemistry over the Amazon.
<https://acp.copernicus.org/articles/18/6461/2018/>

We agree that assessing the performance of the Strat-Trop and CRI-Strat 2 chemical mechanisms, particularly in regions with high biogenic emissions, is very important. To this end, Strat-Trop and CRI-Strat 2 have been thoroughly evaluated against a range of observational data of BVOCs and other important species including surface measurements in the Amazon and Borneo, aircraft campaign measurements and satellite data. These comparisons were published in the Geoscientific Model Development paper Weber et al (2021) (<https://gmd.copernicus.org/articles/14/5239/2021/>) which presented the incorporation of CRI-Strat 2 into UKCA. Indeed, the current study aims to build on the results of Weber et al (2021) by exploring the climatic consequences of the differences in simulated chemistry. While the concise format of Nature Communications prevented us from going into too much detail about this evaluation, we acknowledge that extra detail could be added to provide a clearer indication of the mechanism differences and point the reader to full evaluation. Therefore, we have added the following text to line 165 onwards.

UKESM performance using ST and CS2 was evaluated against present day observational data of BVOCs and other important chemical species from surface sites, flight campaigns and satellites with a full description in Weber et al (2021). Relative to ST, CS2 reduced the model's surface high bias of isoprene and monoterpenes by increasing the local OH concentration. CS2 also yielded substantial improvements in simulated isoprene column over Amazonia, Africa and southeast Asia.

Given the comprehensive evaluation of the ST and CS2 mechanisms in Weber et al (2021), we believe that further mechanism evaluation for this current study is not necessary.

2. While the results of this study are based on preindustrial simulations with a doubling of EBVOC, the conclusions and implications focus on the future. It is important to conduct simulations in the present day and future scenarios as well where BOVC-chemistry-climate interactions are very different than preindustrial conditions due to higher pollution and NOx levels. In my opinion,

preindustrial simulations should not be used to derive conclusions and implications for the future.

The reasons for using a PI atmosphere are also discussed in more detail in response to comment (1) of Reviewer #3.

We agree that background atmospheric composition will be influential in determining how a change to BVOC emissions impacts climate. Indeed, the results of this study support this statement and, in conjunction with the response to comment (1) from Reviewer #3, we have added text to the conclusion to emphasise this to the reader (see below).

We don't argue that an increase in BVOC emissions will have a positive forcing in the present day or future climate. Instead, we state that the result that a doubling of BVOC emissions leads to a positive forcing relates to a PI climate (line 370). The parts of our conclusions which are relevant to the future focus on the argument that determining the climatic impact of BVOC changes, driven by climate change and land use policies such as re/afforestation, will require consideration of the full range of processes by which BVOC emissions influence atmospheric composition. This will include the wide-ranging influence of oxidant changes demonstrated in this work which have been overlooked previously, and the use of a comprehensive mechanism to simulate the chemistry. We believe this conclusion is valid both for the PI atmosphere of this work and present day and future atmospheres and will help inform further work examining the impact of BVOC emission changes in future atmospheres.

Line 396 onwards

.... the critical role of oxidants and sulphate aerosol identified here means the background atmospheric composition, particularly species which affect atmospheric oxidising and background aerosol (e.g., NO_x and SO_x), will be influential in determining how changes to BVOC emissions impact climate.

Doubling BVOC emissions in a present-day or future climate is likely to have a different climatic impact to that simulated here and when assessing the future climatic impact of a re/afforestation policy, contemporaneous background atmospheric composition must also be used. Nevertheless, the processes highlighted in this study provide a framework for such further research.

Minor comment:

2. OH recycling is shown to be important in the CS₂ mechanism affecting the conclusions of this study. But over the Amazon, soil NO_x emissions even (20-50 ppt) could efficiently recycle OH near the surface without the need for OH recycling mechanisms. For example, see the following modeling paper: <https://www.nature.com/articles/s41467-019-08909-4>

AT the very least the role of soil NO_x in OH recycling could be acknowledged.

We agree that the impact of the HO_x-recycling processes in CS₂ do depend on the local chemical environment, particularly concentrations of NO_x. However, we note that other studies have also found that the inclusion of HO_x-recycling has led to significant increases in OH in Amazonia and other major biogenic emission regions including Jenkin et al (2019) and Khan et al (2020) using STOCHEM and, separately, Bates and Jacob (2019) using GEOS-Chem and Novelli et al (2021) using ECHAM-MESSy, highlighting the role these reactions are likely to have on low altitude OH in such regions. Nevertheless, we acknowledge the importance of soil NO_x and have made the following adjustment starting at line 119.

These reactions, along with natural emissions of NO_x from soils, increase simulated OH in environments with high isoprene emissions and low emissions of NO_x from anthropogenic or biomass burning sources, ~~isoprene-rich, NO_x -poor~~ helping to reconcile the persistent model low biases for OH against observations (Bates and Jacob., 2019; Shrivastava et al., 2019; Khan et al., 2020; Weber et al., 2021).

Reviewer #2 (Remarks to the Author):

This paper is clearly written and illustrates an important source of uncertainty in climate modeling that is not generally represented. This work shows how differences in the complexity of chemical mechanisms in an Earth System Model can lead to different climate responses. These differences are due to differences in the oxidation of biogenic compounds (isoprene and terpenes) and resulting gas-phase chemistry, as well as the resulting formation of secondary organic aerosols. The impact of the chemistry of biogenic compounds is quantified by performing simulations with doubled biogenic emissions for both (simpler and complex) chemistry schemes in the UK Earth System Model. The paper describes clearly the important drivers affecting the chemistry effects and climate forcings and feedbacks. Climate models frequently ignore the impacts that atmospheric chemistry has on climate due to the computational cost of including them, therefore this paper is extremely valuable for highlighting the importance of adequately representing the oxidation of some of the most abundant volatile organic compounds in the atmosphere.

I feel the paper could be improved by addressing the following points:

(1) Chemical Mechanisms - it would be helpful to have more information in the main text about the differences between ST and CS2, as these are the basis of most of your figures and results. A table listing the number of species and reactions for different classes of compounds (small HCs, isoprene, terpenes, SOA formation, etc.) for each mechanism would help illustrate the differences.

We acknowledge that the difference between the mechanisms is a key part of this work and have addressed this comment with several alterations. Firstly, in conjunction with the response to Reviewer #1's comment regarding the comparison of ST and CS2 to observations, we have made a clearer reference to the CRI-Strat 2 documentation paper (Weber et al., 2021) which contains a detailed description of the CS2 mechanism.

Secondly, we have amended Figure 1 to include a panel showing the differences between ST and CS2 for the key processes of isoprene oxidation by OH and monoterpene oxidation as well as showing the number of species and reactions in each mechanism. This additional panel is based on original Figure S2 but now also shows that SOA formation in the two mechanisms is the same. As a result, we have removed Fig S2 and renumbered the other figures accordingly. This change has been referenced in the Chemical Mechanisms subsection of Methods (line 457 onwards):

As illustrated in Fig 1(a), ST features isoprene chemistry...

The extra text has also been added to the caption of Figure 1:

(a) Mechanistic differences between ST and CS2 for the key processes of isoprene oxidation by OH and oxidation of monoterpenes (represented by α -pinene and β -pinene in CS2). Processes in black are featured in ST and CS2 while processes in red are only in CS2. RO₂, RCHO and ROOH refer to peroxy radicals, carbonyl and hydroperoxides respectively.

Finally, we have included additional text, from line 459 onwards and shown below, in the Chemical Mechanisms subsection of the Methods section to provide the number of species and reactions in each mechanism as well as the additional emitted species in CS2. Providing a full list in table form would greatly increase the size of the Methods section and we feel this would not be appropriate given the concise format of Nature Communications. Therefore, we have provided a list of the extra emitted species in CS2 and directed to readers to the relevant sections of the mechanism documentation papers. We have also added an additional sentence to the SOA Formation subsection to reiterate the process is the same in ST and CS2.

The Strat-Trop and CRI-Strat 2 chemical mechanisms are described in detail in Archibald et al (2020) and Weber et al (2021) respectively with a full description of every tropospheric chemistry reaction in CS2 also available at <http://cri.york.ac.uk/home.htm> (last accessed 5th June 2022). ST considers 73 species and 305 reactions while CS2 has 228 species and 766 reactions with the bulk of the added complexity coming from a wider range of organic species (Tables 2 and S1 Archer-Nicholls et al., 2021). ST does not feature the CS2 species C₂H₂, C₂H₄, C₃H₆, C₂H₅OH, C₂H₅CHO and methyl ethyl ketone but does add their emissions to species it does consider (e.g., emissions of C₂H₄ are included in C₂H₆ in ST). Some species are omitted entirely by ST and only included in CS2. These are butane, butene, benzene, toluene, oxylene, formic acid and ethanoic acid (Table 3 Archer-Nicholls et al., 2021).

In response to comment (7) of Reviewer #3, we have also added extra detail regarding the treatment of organonitrates in ST and CS (line 457 onwards).

~~and more comprehensive treatment of organonitrates.~~ CS2 also simulates organonitrate formation from a wide range of RO₂ whereas ST uses the methyl nitrate (CH₃ONO₂), isoprene nitrate (C₅H₉NO₃) and nitrooxy aldehyde (C₂H₃NO₄) to represent all organonitrates. As discussed in the main text and Fig 1(a), the oxidation products from α -pinene and β -pinene undergo further chemical reactions which facilitate O₃ and OH production, while the MT species in ST only produces the inert SOA-precursor Sec_Org_{MT} and thus acts as an oxidant sink.

(2) Figure 4 illustrates the feedbacks determined from a model with specified oxidants (a) and with detailed chemistry (b). I think it is a valuable figure and an excellent way to illustrate the complexity of the true atmosphere. However, it seems a bit incongruous as the paper does not use any simulations with specified oxidants (aerosols only). The work presented compares simpler and more complex chemistry, so I would expect such a diagram to illustrate the work done for this paper. Perhaps including a 3rd diagram in the figure showing the ST mechanism, assuming the current panel b represents the CS2 mechanism. This would also address my previous point about showing how the ST and CS2 mechanisms differ.

We are pleased that the reviewer thinks that Figure 4 is valuable. However, we do feel that the comparison between (a) and (b) is valid in the context of this study since a key theme of this study is the importance of chemistry and oxidants on BVOC forcing and how, when they are included in the modelling, we see a much more complex atmospheric response; something illustrated clearly by the comparison of (a) and (b). Panel (a) also relates closely to several prior studies on the forcing from BVOC emissions which have focused solely on aerosol (e.g., Kulmala et al., 2004; Makkonen et al., 2012; Carslaw et al., 2013; Scott et al., 2014; Sporre et al., 2019), and thus places this study more clearly in the context of the wider field.

Since the key processes involved in the response to a doubling of BVOCs are the same in ST and CS2 (it is the magnitude of these processes which differ between the mechanisms), panel (b) represents both mechanisms and we note that Figure 1(b) already clearly illustrates the differences in mechanism response in a forcing context. However, we do acknowledge that we could differentiate between the mechanisms in Figure 4(b) more clearly as well and, to this end, we have modified panel (b) such that the feedback factor for ST and CS2 (α_{ST} and α_{CS2} respectively, defined in the Methods section) are shown for each feedback loop, allowing the reader to see the differences between the mechanisms. We have added an additional sentence to the conclusion to explain this.

(3) A paper that does show the differences in organic aerosols between using specified oxidants and detailed is Tilmes et al., JAMES, 2019. While this is not the primary point of your paper, it might be

worth referencing.

Tilmes, S., Hodzic, A., Emmons, L. K., Mills, M. J., Gettelman, A., Kinnison, D. E., et al. (2019). Climate forcing and trends of organic aerosols in the Community Earth System Model (CESM2). *Journal of Advances in Modeling Earth Systems*, 11. <https://doi.org/10.1029/2019MS001827>

We thank the reviewer for highlighting this paper and we have added it as a reference in the introduction (line 66).

Minor points:

(4) I.117 - give chemical formula for ISOPOO

We have added the chemical formula ($C_5H_9O_3$) to the main text (line 117).

(5) I.151 - 'advanced' in greek letters!

We are not sure what caused this strange change and have corrected the text to read "advanced".

(6) I.178 - define SARF - RF is defined, but what does SA stand for?

SARF stands for stratospheric-temperature adjusted radiative forcing. The fact that this forcing is stratospheric-temperature adjusted is a consequence of the kernel used (Skeie et al., 2020). We used this kernel for consistency with AerChemMIP studies (e.g., Thornhill et al., 2021). We have added the following text to the Forcing Definitions of the Methods subsection (line 542).

The forcing from ozone changes was isolated using the radiative kernel from Skeie et al (2020) as in Thornhill et al (2021) which yielded the stratospheric-temperature adjusted radiative forcing ($SARF_{O_3}$).

(7) Fig.1 caption: need to explain x-axis labels and which y-axis they correspond to.

We have expanded the caption to make it clear that the x axis labels correspond to forcing arising from changes to O_3 , CH_4 , the aerosol DRE and aerosol-cloud interactions. We have also added extra text to make it clear to what the vertical axes correspond. Given the decision to add an extra panel to Figure 1 to resolve this reviewer's request for more detail regarding the mechanistic differences, we have removed the third vertical axis (feedback factor) for clarity.

(b) Forcing and radiative efficiency (ϕ) and feedback factor (α) for the individual forcing components from changes to O_3 ($SARF_{O_3}$), CH_4 , the aerosol DRE (IRF_{DRE}) and aerosol-cloud interactions (CRE) and their combined totals (Net) for ST_{Δ} and $CS2_{\Delta}$. The left axis shows the radiative forcing and the right axis the radiative efficiency.

Data availability: doi link is invalid (or not yet working).

We have finalised the data submission to the repository and the data is now accessible at: <https://doi.org/10.17863/CAM.83526>

Reviewer #3 (Remarks to the Author):

This manuscript describes key differences between two sets of paired of climate simulations designed to test the effects of the models' oxidation mechanisms for biogenic volatile organic compounds (BVOCs) on their simulated climate feedbacks. Huge advances have recently been made in the complexity and accuracy of BVOC oxidation mechanisms, but these are often only incorporated into atmospheric chemistry models; climate and earth-system models still tend to use simple and potentially outdated chemistry schemes. The analysis in this work is therefore a highly important diagnosis of the difference that these new BVOC oxidation mechanisms can make to climate simulations. Furthermore, while most similar studies focus on the effects of BVOC oxidation on secondary organic aerosol (SOA) and ozone formation, the authors of this study notably find that differences in tropospheric oxidation capacity between BVOC mechanisms can have stronger effects on climate, through changes in the methane lifetime and SO₂ oxidation.

The methodology in the manuscript is clear and the simulations and their workup appear well executed. My main complaints (detailed below) are about the remaining simplifications in the more complex chemistry scheme employed here, particularly as they relate to SOA formation. Much about BVOC oxidation and SOA formation remains uncertain and/or is understandably too complicated to incorporate in detail into a model of this scale, and I am not trying to advocate that the authors re-run their simulations with more complex chemistry added, but more discussion and emphasis is needed on the potential for the simplifications in this model to obscure or alter the important feedbacks of BVOC on aerosol formation and thus on climate. Similarly, it would be useful to discuss in more detail the ways in which the particular setup of these simulations -- especially the use of the pre-industrial baseline and the use of a doubling in BVOC emissions to diagnose feedbacks -- might differ from other relevant changes (e.g., marginal changes in BVOC emissions, or present-day conditions), to avoid readers extrapolating these results to conditions in which they might not be strictly applicable.

Specific questions:

(1) L 128: Why are preindustrial conditions chosen as a baseline, and to what extent can these results be extrapolated to changes in present-day BVOC emissions? The effects of BVOC emissions are likely very different in the present-day atmosphere -- in which anthropogenic NO_x increases the ozone production efficiency of BVOCs and anthropogenic SO_x increases the SOA production efficiency of BVOCs -- than they were in the preindustrial atmosphere, and in ways that are strongly affected by the chemical mechanism used (e.g. the treatment of organonitrates, which play a much larger role in the higher-NO_x present-day atmosphere than they did in the preindustrial atmosphere).

(2) Similarly, why is a doubling of BVOC emissions chosen as the perturbation, rather than, say, a marginal change or a zeroing of BVOC emissions? It seems that a zeroing would provide a more precise description of the actual impact of BVOC chemistry, while a marginal change would provide a more realistic baseline for extrapolation to actual likely changes. Because the subsequent chemical outcomes of these perturbations can so often be non-linear, the effects of a global BVOC doubling may differ substantially from marginal effects, or from global reductions in BVOC emissions. (I am not trying to say that the authors need to go back to square one and run new simulations, but rather that some attention should be paid in the manuscript to the consequences of these particular choices and the extent to which these results can be extrapolated).

We have responded to comments (1) and (2) together.

PI conditions were chosen for two reasons. Firstly, the use of a PI atmosphere is crucial to improve the understanding of pristine conditions where there is a high degree of uncertainty and there are a low number of observational constraints. Quantifying the drivers of uncertainty under PI conditions is critical for the wider understanding of anthropogenic radiative forcing, as highlighted by Carslaw et al (2013). Our study has identified the simulation of chemistry as an important source of uncertainty, which has hitherto been neglected in previous research. We have the following text to the conclusion (line 396):

Improving the understanding of the pristine PI atmosphere is important given the large degree of uncertainty in the period and the associated consequences for radiative forcing from the PI to the present day (Carslaw et al., 2013). The use of the PI highlights the importance of simulated chemistry to understanding this period and its response to perturbations.

Secondly, the PI atmosphere provides a relatively simple control atmosphere which can serve as the baseline for future work assessing the role of background atmospheric composition on the impact of BVOC emission changes. As discussed in response to comment (2) from Reviewer #1, we agree that the background atmospheric composition will be influential in determining how a change in BVOC emissions influences climate. Further work is underway involving BVOC emission perturbations in different atmospheres to assess the role of the background atmosphere in response to climate; comparison to these PI simulations will aid understanding into the most important factors of the response. It should also be remembered that NO_x emissions are simulated to decrease from 2020 levels in most future scenarios (Fig 1 Turnock et al., 2020) meaning future atmosphere will tend towards the PI in this regard, making the results of this current study even more relevant.

As discussed in response to Reviewer 1, we have stated in our conclusions that the result of a positive feedback is for a PI atmosphere and have not extrapolated to other atmospheres by making predictions about the forcing from a BVOC emission change in a present day or future atmosphere. Rather we reiterated the influence of the background atmosphere, including references to the importance of SO_x and NO_x, as well as noting that the response to a doubling of emissions is likely to be different in a present day or future atmosphere with the addition of the following text to the conclusion (line 396 onwards):

It [The use of a PI atmosphere] also allows this study to serve as a baseline for future work since the critical role of oxidants and sulphate aerosol identified here means the background atmospheric composition, particularly species which affect atmospheric oxidising and background aerosol (e.g., NO_x and SO_x), will be influential in determining how changes to BVOC emissions impact climate.

Doubling BVOC emissions in a present-day or future climate is likely to have a different climatic impact to that simulated here and when assessing the future climatic impact of a re/afforestation policy, contemporaneous background atmospheric composition must also be used. Nevertheless, the processes highlighted in this study provide a framework for such further research.

We believe these manuscript changes address the reviewer's concern that these results may be extrapolated to situations where they are not as valid by ensuring that readers understand the place of this study and how further work, using future atmospheres, could build on this study.

The use of a doubling of BVOC emissions provides a sufficient signal to overcome model noise. We acknowledge that a doubling of BVOC emissions is a substantial perturbation and the potential non-linearity of the response. However, the uncertainty in (present day) BVOC emissions is already nearly a factor of 2 (e.g., 350 to 670 Tg yr⁻¹ for isoprene; Messina et al., 2016) (and similar, if not larger, uncertainties will exist for PI conditions) and the uncertainties in the key processes (e.g., SOA yield

as Reviewer #3 highlights below) are likely to be greater than the errors introduced by the assumption of linearity. Furthermore, other studies have used greater scaling factors to probe the role of certain forcing agents (e.g., 10x black carbon and sulphate aerosol; Sherman et al., 2021), suggesting our approach is not an outlier. We have added the following text to the conclusion (from line 396):

Doubling BVOC emissions represents a substantial perturbation, and this analysis assumes a linear response to BVOC emission changes while some of the atmospheric responses may exhibit non-linearity. However, the uncertainty in BVOC emissions is already substantial (350 to 650 Tg yr⁻¹ for isoprene in the present day; Messina et al., 2016), illustrating a doubling is not so unrealistic, and the uncertainties in key processes (e.g. SOA formation) are likely to exceed the errors caused by the assumption of non-linearity.

(3) L 151: "advanced" should be in Latin not Greek alphabet :)
We have corrected this – not sure how it happened!

(4) L 162-163: *What are these 180 TgC of extra emitted reactive compounds? Do directly emitted non-(isoprene+MT) compounds contribute to the changes between ST and CS2, and if so, how much and what compounds? Also, it is stated here that the higher VOC emissions in CS2 leads to greater OH, while below (L 197) it is stated that an increase in VOC emissions depletes OH. Is the greater OH in CS2 really attributable to the 180 TgC of extra emitted VOCs, or to mechanistic differences?*

On this issue, we discovered an error in our calculations. The 180 TgC yr⁻¹ is the present-day value whereas the PI value is 125 TgC yr⁻¹. The PI value is smaller because many of the species emitted only in CS2 have substantial anthropogenic sources but smaller biogenic sources so the total extra emissions of the species is smaller in the PI. We have corrected the value in the text but do not believe it alters the conclusions.

Most of the extra reactive organic carbon (ROC) in the PI comes from the chemical treatment of monoterpene oxidation products in CS2 compared to ST where monoterpene oxidation only produces the chemically inert Sec_Org_{MT}. We have re-emphasised the difference with the following change to line 155 onwards.

For monoterpenes ST features a single tracer (MT) whose oxidation by O₃, OH and NO₃ produces only a chemically-inert species, Sec_Org_{MT}, which condenses onto aerosol or nucleates new aerosol with sulfuric acid. This lack of further chemistry means MT only acts as an oxidant sink rather than behaving as reactive organic carbon (ROC) (Heald and Kroll; 2020).

In response to comment (2) of Reviewer #2 requesting greater information regarding mechanism differences, we have added further information to the Chemical Mechanisms subsection of Methods. We have included a list of the species emitted in CS2 and not in ST but where the mass of these species are added to existing ST emissions (e.g., emissions of C₂H₄ are included in the emissions of C₂H₆ in ST) and, separately, a list of species which are only in CS2 and ignored entirely by ST (as well as directing the reader to original documentation). Only the latter list contributes to the total extra VOC emissions in CS2 and, relative to the extra VOC emissions from the treatment of monoterpenes, is small.

Regarding oxidant depletion, increasing monoterpene emissions will deplete oxidants in both mechanisms via the direct reaction of the monoterpene species with OH, O₃ and NO₃. In ST this is all

that happens from a chemical perspective and thus MT only acts as an oxidant sink. However, in CS2, MT (as α -pinene and β -pinene) acts initially as an oxidant sink but also subsequently as an oxidant source via the degradation of the resulting oxidation products (e.g., photolysis of carbonyls and peroxides), which can offset, partially or fully, the original depletion. UKESM1 simulations performed using the CRI-Strat chemical mechanism (an earlier version of CS2) with its standard VOC emissions (the same as used by CS2) and VOC emissions from Strat-Trop found that the greater VOC emissions used in CRI-Strat and CS2 led to lower OH at the surface (due to greater OH depletion by reaction with VOCs) but higher OH in the tropical lower free troposphere (Fig 9 Archer-Nicholls et al., 2021).

We acknowledge that the spatial variation in oxidant change could be made clearer and have made the following amendment from line 161.

~~The chemical treatment of monoterpenes and the wider range of VOCs considered by CS2 means CS2's emissions of reactive organic carbon are ~180 TgC yr⁻¹ (17.5%) higher than ST, leading to greater OH in the tropical planetary boundary layer (PBL; lowest ~1-2 km) and lower free troposphere (FT) (Archer-Nicholls et al., 2021).~~ ~~In the PI, CS2 simulates an extra ~125 TgC yr⁻¹ of reactive organic carbon emissions than ST, primarily due to the chemical treatment of monoterpenes with a smaller contribution from the extra emitted species considered by CS2 (Methods). Prior mechanistic analysis has identified this additional ROC to lead to lower surface OH but greater OH in the tropical lower free troposphere (FT) (Archer-Nicholls et al., 2021).~~

(5) L 203: "greater increase OH production" should probably have an "in" in there?
"in" has been added.

(6) L 204: Why does CS2 have higher OH production from hydroperoxide (ROOH) photolysis than ST? The reasoning for the increase OH production from HO₂+NO is well explained in L 206-210, but the ROOH contribution isn't described.

In the region referenced in this sentence (lower tropical FT) the higher OH production from ROOH production in CS2 comes predominantly from the greater increase in ROOH produced from α -pinene and β -pinene (e.g., RCOOH₂₅ and RTN₂₈OOH). This was isolated with short sensitivity tests where the isoprene chemistry in CS2 was reverted to that in ST. We have added the red text below to make this clearer (on line 204).

and hydroperoxide (ROOH) photolysis, primarily coming from the ROOH derived from α -pinene and β -pinene (omitted in ST), in CS2_Δ.

(7) L 234-242: It is surprising that PAN-driven changes in NO_x transport between the two mechanisms dominates over the effects of NO_x depletion from organonitrates. To what extent do the mechanisms' treatments of organonitrates differ, and to what extent do both the organonitrates and their differences between the mechanisms contribute to BVOC-driven changes in ozone formation? If the CS2 mechanism included rapid aqueous hydrolysis of tertiary nitrates (Vasquez et al., 2020; DOI: 10.1073/pnas.2017442117), would the conclusions change?

ST uses the methyl nitrate (CH₃ONO₂), isoprene nitrate (C₅H₉NO₃) and nitrooxy aldehyde (C₂H₃NO₄) to represent all organonitrates (a comment we have added to the Chemical Mechanism subsection of Methods in response to Reviewer #2 request for more mechanism information). CS2 forms organonitrates from a wide range of RO₂ including those derived directly and indirectly from isoprene and monoterpenes. Despite this, the total RONO₂ burden is only 4% higher in CS2 than ST.

The tropospheric burden of organonitrates in the base runs was ~1.3% of total organic nitrated species (NO_y) and this increased to ~1.7% of NO_y in the 2xBVOC in both mechanisms. This response was much smaller in absolute terms than the change to PANs (15% increased to 23% in ST; 10% increased to 15% in CS2) and HONO_2 (52% decreased to 47% in ST; 58% increased to 54% in CS2), hence our focus on PANs and HONO_2 . However, we acknowledge that the role of organonitrates would probably be higher in present day or future atmospheres with higher anthropogenic emissions of NO_x and VOCs. ST and CS2 simulated greater fractions of organonitrates as a total fraction of NO_y in previous studies using present day emissions (Archer-Nicholls et al., 2021; Weber et al., 2021), further highlighting the role that background atmospheric composition is likely to play in the response to a change in BVOC emissions.

At present there is no aqueous phase loss of organonitrates in either mechanism; organonitrates are lost via wet and dry deposition, photolysis and reaction with OH. While including this aqueous phase loss would be an improvement and probably influential for atmospheres with higher NO_x emissions such as the present day or certain future scenarios, the smaller role that organonitrates play in the PI in terms of NO_x sequestration and O_3 production means it is likely to have a smaller effect.

(8) L 330: "Typically" appears twice.

The second "typically" has been removed.

(9) L 492-496: *While SOA formation schemes in complex models such as these must necessarily be simplified, the simplifications here, employed even in the updated CS2 mechanism, seem potentially problematic in ways that could influence the paper's conclusions:*

(1) *SOA precursors are not produced in fixed yields, but rather depend on the fates of peroxy radicals in the BVOC oxidation cascade. For example, the dominant SOA precursor from isoprene is presumed to be IEPOX, formed in the $\text{RO}_2 + \text{HO}_2$ pathway, while dimerization from $\text{RO}_2 + \text{RO}_2$ reactions is thought to be the most efficient SOA-forming pathway for terpenes. Thus, the yields of SOA precursors should be allowed to vary with the BVOC oxidation pathways; using a fixed yield instead shuts off this potential avenue for changes in SOA yields. (Furthermore, SOA precursors are not inert, as represented here, but themselves have oxidation pathways that may limit their contribution to aerosol in regions with higher oxidative capacity or lower preexisting aerosol loadings).*

(2) *SOA production from potential precursors is often not a given, and not condensation driven, but rather dependent upon other aerosol conditions, especially for reactive uptake. For example, IEPOX is not guaranteed to produce SOA, and its SOA formation is reactive, not condensation-driven. This reactivity depends on aerosol liquid water content, acidity, sulfate content (both anthropogenically driven) and organic shell (see, e.g., Nah et al. 2019, DOI: 10.1016/j.atmosenv.2019.116953 and Gaston et al. 2014, DOI: 10.1021/es5034266). Each of these dependencies can provide self-limiting feedbacks both from the extent of SOA formation and from other features of BVOC oxidation (e.g. the changes in SO_2 fate highlighted in this manuscript).*

(3) *Relatedly, the SOA yields used here may not apply to preindustrial conditions; for example, a 3% yield from isoprene was found to fit observations in the present-day Southeast US well, but this is a region with relatively acidic, sulfate-rich aerosol that would probably lead to higher IEPOX uptake than in most preindustrial environments (Marais et al. 2016, DOI: 10.5194/acp-16-1603-2016).*

(4) *A key part of the conclusions in this manuscript depends on the timescale by which SOA is produced following BVOC emission, because lower OH in ST leads to slower BVOC oxidation and a greater permeation of SOA precursors to the free troposphere (L 251-255). However, this requires*

that the timescale of SOA formation be correctly calibrated to begin with. It isn't clear from the condensation reactions listed here what timescale is employed in these simulations, but one certainty is that IEPOX formation, and thus SOA formation from isoprene, is not a one-step oxidation process as suggested here; instead, it requires two generations of OH oxidation, which may make a substantial difference to the fraction of IEPOX that reaches the free troposphere relative to the inert precursor represented here.

(5) Whether or not SOA formation is condensation-driven, it is not an irreversible process, as it seems to be in these simulations; dilution of air masses in which SOA formed can cause the revolatilization of SOA driven by condensation equilibria, and condensed-phase or heterogeneous reactions of aerosols can cause fragmentation of organic SOA constituents.

I know this is a long list of complaints about the SOA schemes here -- again, I'm not trying to say the authors should change and redo their entire simulations, but much more nuanced discussion is required of the uncertainties and potential for additional factors to influence BVOC SOA formation and its dependencies.

We thank the author for the helpful comments on SOA and agree that improving the description of SOA formation and growth in Earth System models, and especially UKESM1, is a priority.

Rigorous modelling of aerosol-chemistry-oxidant coupling on a global scale, and the investigation of its links to climate, is still a relatively new field. Many of the other in-depth studies on the subject have used a similar approach to UKESM with SOA formed via condensation of inert SOA-precursors. For example, Sporre et al (2020) considered three climate models (ECHAM, EC-Earth and NorESM) where SOA formation from isoprene and monoterpenes occurred via the condensation of inert SOA-precursors (produced from a single oxidation step of isoprene and monoterpenes) on to existing aerosol and various aerosol nucleation pathways (sulfuric acid only, pure biogenic nucleation and co-nucleation of certain SOA-precursors with sulfuric acid). Karset et al (2018) used a similar approach. Therefore UKESM, in the setup used in this study, is comparable to some of the most relevant recent aerosol-oxidant coupling studies. Nevertheless, we acknowledge there is room for improvement.

Given the uncertainty surrounding the modelling of IEPOX uptake, the fact that IEPOX is not included in Strat-Trop, and the lack of a functional terpene-dimer scheme, we decided to use a consistent SOA formation approach for this work while also making efforts to improve it beyond the standard setup used for AerChemMIP by rewriting the chemistry and aerosol code in UKESM to include SOA formation from isoprene. This approach allowed us to constrain the difference in aerosol response to that arising solely from the chemistry scheme (and not the simulation of aerosol formation). However, we acknowledge that this approach will not capture the full aerosol-chemistry-oxidant coupling, due to points raised by Reviewer #3, particularly the issue of the time scale of SOA-precursor production (and its dispersion) and the dependence of reactive uptake on aerosol composition. Indeed, simulating SOA-precursor production from later generation oxidation products will increase the dependence on oxidant concentrations for the production of SOA and thus most likely amplify the impact of simulated oxidant changes driven by differences in chemical mechanism or external factors like anthropogenic emissions.

Nevertheless, we believe that the CS2 simulations, given the more sophisticated chemistry, provide a good baseline for the future evaluation of the impact of the key improvements mentioned by Reviewer #3, specifically IEPOX uptake, terpene dimer formation, and revolatilisation of SOA.

To address the concerns of Reviewer #3 we have added the following text along with other amendments to the conclusion (from line 396 onwards).

Improvements to the description of SOA formation beyond the current fixed yield, condensation-driven approach via the adoption of more realistic processes including dimer formation from terpenes (Weber et al., 2020) and the reactive uptake of isoprene epoxy-diols (IEPOX) (Gaston et al., 2014) are likely to be impactful and open additional feedback loops, warranting further work. The requirement for multiple oxidation steps for SOA-precursor formation will alter (and indeed likely accentuate) the effect of oxidants on SOA lifetime while the complex role of NO_x in IEPOX and dimer formation and the influence of aerosol composition (e.g. acidity) on IEPOX reactive uptake will drive a greater dependence on background atmospheric composition.

We believe this addition makes clear to the reader the requirement for further improvements to SOA simulation and how they will drive a greater dependence on atmospheric composition and oxidant concentrations.

References

K.H Bates and D.J Jacob. A new model mechanism for atmospheric oxidation of isoprene: global effects on oxidants, nitrogen oxides, organic products, and secondary organic aerosol. *Atmospheric Chemistry and Physics*, 19(14), 9613-9640 (2019).thorn

K. S. Carslaw et al. Large contribution of natural aerosols to uncertainty in indirect forcing. *Nature*, 503, 7474, 67-71 (2013).

M.A.H. Khan et al. Changes to simulated global atmospheric composition resulting from recent revisions to isoprene oxidation chemistry. *Atmospheric Environment* 244, 117914 (2021).

M.E. Jenkin et al. The CRI v2. 2 reduced degradation scheme for isoprene. *Atmospheric Environment* 212, 172-182 (2019).

A. Novelli et al. Importance of isomerization reactions for OH radical regeneration from the photo-oxidation of isoprene investigated in the atmospheric simulation chamber SAPHIR, *Atmos. Chem. Phys.*, 20, 3333–3355, 2020.

P. Sherman et al. Sensitivity of modeled Indian monsoon to Chinese and Indian aerosol emissions. *Atmospheric Chemistry and Physics*, 21(5), pp.3593-3605 (2021).

R.B. Skeie et al. Historical total ozone radiative forcing derived from CMIP6 simulations. *Npj Climate and Atmospheric Science* 3.1,1-10 (2020).

G. Thornhill et al. Climate-driven chemistry and aerosol feedbacks in CMIP6 Earth system models. *Atmospheric Chemistry and Physics* 21.2, 1105-1126 (2021).

S. T. Turnock et al. Historical and future changes in air pollutants from CMIP6 models, *Atmos. Chem. Phys.*, 20, 14547–14579 (2020).

REVIEWER COMMENTS

Reviewer #1 (Remarks to the Author):

The authors have acknowledged caveats in their work mainly with limitations in using just PI background, simple SOA schemes etc. I agree with the authors that using complex chemistry mechanisms like CS₂ are important compared to schemes that use prescribed oxidants or ST.

Still I am worried that the results in Figure 1b are misleading since they imply a strong positive forcing from doubling of BVOC emissions. Probably they should call it responses to changes in BVOCs in PI atmosphere, rather than forcing. In my understanding, the forcing in climate modeling is almost always calculated as a difference between preindustrial and present day scenarios. Figure 1b is difficult to interpret in that sense.

Would this general result stand regardless of SOA mechanisms, and changing SOA yields in present day conditions?

Also role of SOA on cloud forcing is not fully considered here mainly because of simplistic and low SOA yields, absence in consideration of SOA interactions with NO_x and SO₂, and lack of treatments of SOA cloud chemistry that may be important in large portions of the marine atmosphere.

I agree that the authors provide a solid gas-phase chemical modeling framework with CS₂ and present its value for the global modeling community . Still there are several unresolved questions as I highlight above. Mainly it needs to be seen in the context of differences between present day and preindustrial atmosphere, if increases in CH₄ and O₃ indeed cause a net positive response to changes in BVOCs compared to SOA formation and its interactions with cloud chemistry.

Reviewer #2 (Remarks to the Author):

I feel the authors have satisfactorily addressed my previous comments and that the paper is worthy of publication.

Reviewer #3 (Remarks to the Author):

Through their additions to the manuscript, the authors have largely addressed the issues brought up in the initial review. In particular, the added discussion of uncertainties and potential bias inherent in this linearized approach (especially to SOA formation) are helpful, as is the fact that the authors highlight this as a framework for further radiative forcing analysis rather than a precise calculation of the radiative response of a doubling of biogenic emissions.

A few points remain to be addressed:

Regarding the use of a doubling as the perturbation from PI:

In the final sentence added to the conclusion, it's an assumption of linearity, not an assumption of non-linearity.

Furthermore, it's not particularly correct to say that " this analysis assumes a linear response to BVOC emission changes". For some cases (e.g., SOA formation, with its fixed yield) this is precisely true, but for other cases the mechanism accounts for some non-linearity (e.g. OH changes), and it would only be in extrapolating linearly from the emission doubling to other emission changes that an assumption of linearity would be introduced. Thus, some of the radiative forcing responses shown in Figure 1 could be expected to behave linearly with emissions in the modeling framework used here (even if not in reality), while others would not, and the extent of their non-linearity is not known. This distinction seems worth mentioning.

The factor of 2 uncertainty on isoprene emissions doesn't really represent an equivalent uncertainty on many outcomes, because model parameters are often tuned to their preset conditions -- e.g., the SOA yield from isoprene would have to be scaled down if emissions were really doubled in the model, because it is tuned, to an extent, to the preset isoprene emissions. Either way, bringing up this uncertainty doesn't help to calibrate the reader to the potential magnitude of non-linearities, it just accentuates the uncertainties inherent to this modeling.

Regarding the updated discussion of the differences in reactive organic carbon (ROC) emissions between the two mechanisms:

The inclusion of greater detail about which VOCs are explicitly represented in each mechanism is very helpful, but the provenance of the 125 TgC/y of extra ROC in CS2 remains unclear -- it is stated that these ROC are "emitted" but then that they are attributable to the treatment of monoterpene products. Presumably the same amounts of monoterpene are *emitted* in each mechanism, though, right, and the reactive monoterpene products are not themselves emitted? Are the reactive monoterpene products being counted as emissions? It would probably be most helpful to split these distinct ROC sources up and tell the reader separately (a) how much greater the total organic emissions are in CS2 due to the inclusion of newly emitted VOCs, and (b) how much the new monoterpene chemistry in CS2 contributes to additional ROC. (Reading this over again, it seems the confusion stems more from the definition of "reactive organic carbon"; the fact that monoterpenes react with oxidants in ST means that despite their lack of chemically active products, they are themselves ROC).

Response to Reviewers

We are grateful to the reviewers for their comments and efforts which have helped us improve this manuscript. We have responded to each reviewers' comments sequentially below with italicised text showing the reviewer's comments and plain text showing our response. Text which has been added to the manuscript is coloured red. Current manuscript text is in blue and any text which has been removed from the manuscript is blue and has been struck through. The locations of changes are stated (line number refer to the numbering in the updated, marked-up manuscript) . Where reviewer comments were unnumbered, we have added numbers for ease of reference, e.g. (1). We hope these revisions address the concerns of the reviewers.

Reviewer #1 (Remarks to the Author):

The authors have acknowledged caveats in their work mainly with limitations in using just PI background, simple SOA schemes etc. I agree with the authors that using complex chemistry mechanisms like CS₂ are important compared to schemes that use prescribed oxidants or ST.

(1) Still I am worried that the results in Figure 1b are misleading since they imply a strong positive forcing from doubling of BVOC emissions. Probably they should call it responses to changes in BVOCs in PI atmosphere, rather than forcing. In my understanding, the forcing in climate modeling is almost always calculated as a difference between preindustrial and present day scenarios. Figure 1b is difficult to interpret in that sense.

While we agree that radiative forcing can be discussed in terms of the forcing of change from the pre-industrial (PI) to present day (PD), radiative forcing can be calculated between any two climate simulations since it is the "net change in the energy balance of the Earth due to some imposed perturbation" (Section 8.1.1, IPCC AR5, Myhre et al., 2013). It is common practice to change just one component of the system to isolate its impact, as opposed to the multiple radiative-relevant changes which have occurred between the PI and PD (e.g., GHGs, aerosol, land use/surface albedo). For natural emissions, such as BVOCs, it makes less sense to examine the impact of the PI to present day changes since these are driven by a range of factors (temperature, land use change, parameterisation of the impact of CO₂ and temperature on BVOC emissions, etc) and the disagreement in these between models leads to a range of PI to PD emission changes. The conventional approach, widely employed in the literature and in particular by the AerChemMIP part of CMIP6 (Collins et al., 2017) and the IPCC AR6 report (Section 7.4.2.5.1 Forster et al., 2021), is a doubling of a natural emissions, and this is what has been done in this study. This doubling is then used to calculate the forcing per unit change in emission (radiative efficiency) and the feedback factor which quantifies the strength of the feedback. (The feedback factor is defined in detail in the Methods section.)

We make it clear in the abstract and manuscript (line 130 onwards) that we are considering the radiative forcing following a doubling of BVOC emissions in a PI atmosphere, not the PI to present day change in BVOC emissions. Nevertheless, we agree Figure 1(b) could be made clearer. We have kept the term radiative forcing, since it is valid to refer to the radiative forcing arising from a doubling of BVOC emissions, but to address the reviewer's concerns we have updated the title and caption as follows to make it clear to the reader that the radiative forcing refers to a doubling of BVOC emissions in a PI atmosphere, not PI to present day. Please also note that we have decided to keep the third axis showing the feedback factor in Figure 1(b) since the feedback is discussed in the conclusion.

Title

~~(b) Forcing, Radiative Efficiencies (ϕ)~~

(b) Radiative Forcing, Radiative Efficiency (ϕ) and Feedback Factor (α) calculated from a doubling of BVOC Emissions in a PI Atmosphere

Caption

(b) Radiative forcing, ~~and~~ radiative efficiency (ϕ) and feedback factor (α) from a doubling of BVOC emissions in a PI atmosphere. ~~for~~ We show the individual forcing components from changes to O₃ (SARF_{O3}), CH₄, the aerosol DRE (IRF_{DRE}) and aerosol-cloud interactions (CRE) and their combined totals (Net) for ST_Δ and CS_{2Δ}. The

left axis shows the radiative forcing, and the inner right axis the radiative efficiency and the outer right axis the feedback factor. Error bars in (b) show the standard error.

The key message from Figure 1(b) is that radiative forcings (or radiative efficiency or feedback factor) for each component and the net values are different between the mechanisms and we believe this is clearly shown.

(2) Would this general result stand regardless of SOA mechanisms, and changing SOA yields in present day conditions?

(3) Also role of SOA on cloud forcing is not fully considered here mainly because of simplistic and low SOA yields, absence in consideration of SOA interactions with NO_x and SO₂, and lack of treatments of SOA cloud chemistry that may be important in large portions of the marine atmosphere.

We have responded to (2) and (3) together below.

For (2), we acknowledge there are two points to consider. Firstly, whether changing the SOA mechanism would change the response in the PI atmosphere considered and, secondly, how the different atmospheric conditions (and by extension different SOA yields) of the present day would affect the response.

Following comments from Reviewer 1 and Reviewer 3 in the previous round of review regarding the importance of the background atmosphere, we emphasised in the conclusion that the radiative forcing arising from a doubling of BVOC emissions is likely to be different in a present day or future atmosphere (line 446). (The sentence below is amended further in response to comment (4) but the conclusion is preserved.)

Doubling BVOC emissions in a present-day or future climate is likely to have a different climatic impact to that simulated here...

This statement is based on this study's findings of the importance of oxidants and sulphate aerosol which will be different in present day or future atmospheres. The following text (line 425 onwards), added in the response to the previous round of reviewer comments, and the additions in red made in this round highlight this:

... the critical role of oxidants and sulphate aerosol identified here means the background atmospheric composition, particularly species which affect atmospheric oxidising capacity and background aerosol (e.g., NO_x and SO_x which are higher in the present day than the PI), will be influential in determining how changes to BVOC emissions impact climate.

In response to the previous set of reviewer comments, particularly those from Reviewer 3, we also added a section to the conclusion pertaining to aerosol (line 430 onwards) indicating the limitations of the current approach SOA simulation and, the additional processes, which might be beneficial but are not currently widely incorporated in climate models. We stressed that these additions would likely accentuate the dependence of the response on the background atmosphere, which we noted involved NO_x and SO_x.

In response to comment (3) of this review, we have added further text noting the current omission of SOA formation in aqueous aerosol and cloud droplets and its importance. In response to comment (2) of this review, we have also highlighted that these SOA updates may change the response of the DRE and ACI to a BVOC emission perturbation (i.e., an updated SOA mechanism may change the response). In the next sentence we have added further text stating that the response of DRE and ACI will be influenced by background atmospheric composition (attending to the second part of (2)). We already mention the dependence on NO_x and have added a reference to SO_x as well attending to comment (3).

Improvements to the description of SOA formation beyond the current fixed yield, condensation-driven approach ~~via~~ include the adoption of more realistic processes including dimer formation from terpenes (e.g., Weber et al., 2020), the reactive uptake of isoprene epoxy-diols (IEPOX) (Gaston et al., 2014), and SOA formation in aqueous aerosol and cloud droplets which is believed to be comparable to gas phase SOA formation in some circumstances (e.g. Ervens et al., 2011). These additions are also likely to be impactful and open additional feedback loops, may alter the DRE and ACI responses to a BVOC emission perturbation, but

are not widely incorporated at present in climate models. The response of the DRE and ACI will be influenced by background atmospheric composition and the requirement for multiple oxidation steps for SOA-precursor formation will alter (and indeed likely accentuate) the effect of oxidants on SOA dispersion and lifetime while the complex role of NO_x in IEPOX and dimer formation and the influence of aerosol composition (e.g., SO_x and acidity) on IEPOX reactive uptake so will also drive a greater dependence on background atmospheric composition.

We also note that the PI atmosphere serves as providing a baseline for future work (line 424):

It [use of the PI] also allows this study to serve as a baseline for future work...

Overall, we believe these sections of text, including the recent additions in response to comments (2) and (3), provide readers with a clear illustration of the scope of the current work, its limitations such as the relatively simplistic approach to SOA modelling (and how more advanced approaches may alter the response) and the dependence on background atmospheric composition, while identifying the areas where future work would be most useful. We also note the additions mentioned are not widely simulated in climate models and so assessing their impact would require significant model development and evaluation which is beyond the scope of this study.

(4) I agree that the authors provide a solid gas-phase chemical modeling framework with CS₂ and present its value for the global modeling community. Still there are several unresolved questions as I highlight above. Mainly it needs to be seen in the context of differences between present day and preindustrial atmosphere, if increases in CH₄ and O₃ indeed cause a net positive response to changes in BVOCs compared to SOA formation and its interactions with cloud chemistry.

We thank the reviewer for acknowledging that our study provides value for the global modelling community with its solid gas-phase chemical modelling framework. However, we respectfully disagree that the results of this study need to be seen in the context of the difference between the PI and the present day for reasons discussed in our response to comment (1). Performing further simulations at considerable cost (given the multi-decadal simulations required) using present day conditions is beyond the scope of this study.

The study shows that in a PI atmosphere in UKESM1 the radiative forcing arising from a doubling of BVOC emissions is positive when two different chemical mechanisms are used but that the magnitude of forcing is substantially different between the mechanisms. This is an important result and, by focusing on a single time period, we have been able to discuss the drivers of this difference in detail and thus illustrate the key role of chemistry and oxidants in determining the climatic impact of a BVOC emission change, including involvement in processes not previously considered in the context of BVOCs. Following comments from reviewers we have amended the manuscript so that readers will understand that while the quantitative results presented here are applicable for a PI atmosphere (i.e. a doubling of BVOC emissions in a PI atmosphere causes a positive RF in UKESM1) and that the radiative response will likely differ if the same BVOC emission perturbation were performed in a present day atmosphere, readers will also understand the drivers of this difference (e.g., different oxidising capacity, NO_x and sulphate aerosol) and how the processes and framework discussed in this study will allow future work to assess the impact of BVOC emission changes in different atmospheres.

To address the point about the balance between SOA/ACI and O₃ and CH₄ in comment (4) and make it clear how that future work should focus on changes to these components, the following addition has been made to the conclusion (line 424 onwards).

It also allows this study to serve as a baseline for future work since the critical role of oxidants and sulphate aerosol identified here means the background atmospheric composition, particularly species which affect atmospheric oxidising capacity and background aerosol (e.g., NO_x and SO_x which are higher in the present day than the PI), will be influential in determining how changes to BVOC emissions impact will affect O₃, CH₄, aerosol burdens and CDNC and the magnitude of the opposing radiative effects which ultimately determine the climatic impact.

We have also added extra text to abstract to highlight the dependence on background atmospheric composition.

This illustrates the significant influence of chemistry and oxidants on gas and aerosol responses to BVOC emission changes, ~~and the more complex pathways by which BVOCs influence climate than are currently recognised,~~ and the likely dependence on background atmospheric composition.

The fact that the response will likely be different in a present day or future atmosphere does not diminish the importance of this study. We adopted the approach of a doubling of emissions in a PI atmosphere (rather than a PI to PD change) since this is the conventional approach used for natural emissions used in AerChemMIP /CMIP6 and the IPCC AR6 report. This means this study is directly comparable to 2xBVOC emission experiments performed for AerChemMIP. This was a deliberate decision since we plan to compare the impact of different chemical mechanisms on the RF in a single model with the magnitude of the variation in RF between models using the AerChemMIP 2xBVOC experiments and the simulations performed here, as part of future work. Previous studies examining the climatic impact of BVOCs have typically been dominated by the aerosol effect and it has been assumed the RF from a given perturbation would be generally independent of background atmospheric composition (i.e., similar in PI and present day). Little consideration has been given impact of background atmospheric composition, yet we have shown the importance of oxidants and chemistry in (e.g., via the oxidant-driven suppression SO_2+OH and CDNC) and therefore an intrinsic sensitivity to atmospheric composition and description of chemistry.

Thus, the results of this study challenge the conventional approach used in CMIP6 and the IPCC and highlight the fundamental importance of assessing the impact of BVOC emission changes in atmospheric context in which they occur (be it PI, present day, or a future scenario) and provide a framework for this analysis.

This was already referenced in the conclusion (line 413 onwards):

By comparing the response to an E_{BVOC} increase with two interactive chemical mechanisms, this study progresses beyond prior studies by identifying the wider reach of oxidants as they impact not only the forcing from gas phase composition changes but also the forcing from aerosol and cloud property changes; previously overlooked interactions.

However, to emphasise this we have added further text to the conclusion:

The wide-ranging influence of oxidants and chemistry identified in this study, and the attendant dependence on atmospheric chemical composition, means a doubling BVOC emissions in a present-day or future climate is likely to have a different climatic impact to that simulated here. ~~and w~~When assessing the future climatic impact of a re/afforestation policy the application of the radiative efficiency or feedback factor determined using the doubling of emissions in a PI atmosphere following the CMIP6 convention may not suitable. Instead, contemporaneous background atmospheric composition must also be used. ~~Nevertheless,~~ with the processes highlighted in this study providing a framework for such further research.

The conclusion and these amendments provide readers with a clear understanding of the scope and findings of this study, including the important challenge to the CMIP6 convention.

Reviewer #2 (Remarks to the Author):

I feel the authors have satisfactorily addressed my previous comments and that the paper is worthy of publication.

We are pleased that Reviewer #2 is happy with our amendments.

Reviewer #3 (Remarks to the Author):

Through their additions to the manuscript, the authors have largely addressed the issues brought up in the

initial review. In particular, the added discussion of uncertainties and potential bias inherent in this linearized approach (especially to SOA formation) are helpful, as is the fact that the authors highlight this as a framework for further radiative forcing analysis rather than a precise calculation of the radiative response of a doubling of biogenic emissions.

A few points remain to be addressed:

Regarding the use of a doubling as the perturbation from PI:

(1) In the final sentence added to the conclusion, it's an assumption of linearity, not an assumption of non-linearity.

This was an error and should have been "linearity". However, in responses to comments (2) and (3) below, the section where this sentence appears has been removed.

(2) Furthermore, it's not particularly correct to say that "this analysis assumes a linear response to BVOC emission changes". For some cases (e.g., SOA formation, with its fixed yield) this is precisely true, but for other cases the mechanism accounts for some non-linearity (e.g. OH changes), and it would only be in extrapolating linearly from the emission doubling to other emission changes that an assumption of linearity would be introduced. Thus, some of the radiative forcing responses shown in Figure 1 could be expected to behave linearly with emissions in the modeling framework used here (even if not in reality), while others would not, and the extent of their non-linearity is not known. This distinction seems worth mentioning.

We acknowledge the point made in this comment and have amended the corresponding paragraph as follows to emphasise the varying degrees of linearity in the response of the different components and the inherent uncertainty.

Doubling BVOC emissions represents a substantial perturbation, and extrapolation of this study's results to different emission scalings (e.g., 50% increase) should be performed with care since different components of the model's response are likely to scale with emissions with varying degrees of linearity. For example, the current use of a fixed SOA yield means the modelled IRF_{DRE} may scale quite linearly with emissions but the non-linearity of O_x-NO_x-VOC chemistry (e.g., Jenkin and Clemitshaw., 2000) means changes to OH, and thus to CH₄ forcing, are likely to be less linear. The complexity of the interactions and role of background atmospheric composition mean the extent of linearity can only truly be determined by further experiments. ~~and this analysis assumes a linear response to BVOC emission changes (as in Thornhill et al., 2021) while some of the atmospheric responses may exhibit non-linearity. However, the uncertainty in BVOC emissions is already substantial (350 to 650 Tg yr⁻¹ for isoprene in the present day; Messina et al., 2016), illustrating a doubling is not so unrealistic, and the uncertainties in key processes (e.g., SOA formation) are likely to exceed the errors caused by the assumption of non-linearity.~~

(3) The factor of 2 uncertainty on isoprene emissions doesn't really represent an equivalent uncertainty on many outcomes, because model parameters are often tuned to their preset conditions -- e.g., the SOA yield from isoprene would have to be scaled down if emissions were really doubled in the model, because it is tuned, to an extent, to the preset isoprene emissions. Either way, bringing up this uncertainty doesn't help to calibrate the reader to the potential magnitude of non-linearities, it just accentuates the uncertainties inherent to this modeling.

The SOA yields used in this study are based on experimentally determined values rather than being tuned such that the emissions of BVOCs yield an acceptable SOA burden. Nevertheless, we accept the reviewer's point and have removed this section.

(3) Regarding the updated discussion of the differences in reactive organic carbon (ROC) emissions between the two mechanisms:

The inclusion of greater detail about which VOCs are explicitly represented in each mechanism is very helpful, but the provenance of the 125 TgC/y of extra ROC in CS₂ remains unclear -- it is stated that these ROC are

*"emitted" but then that they are attributable to the treatment of monoterpene products. Presumably the same amounts of monoterpene are *emitted* in each mechanism, though, right, and the reactive monoterpene products are not themselves emitted? Are the reactive monoterpene products being counted as emissions? It would probably be most helpful to split these distinct ROC sources up and tell the reader separately (a) how much greater the total organic emissions are in CS2 due to the inclusion of newly emitted VOCs, and (b) how much the new monoterpene chemistry in CS2 contributes to additional ROC. (Reading this over again, it seems the confusion stems more from the definition of "reactive organic carbon"; the fact that monoterpenes react with oxidants in ST means that despite their lack of chemically active products, they are themselves ROC).*

It is correct that the same quantity of monoterpenes is emitted by each mechanism and the oxidation products are not emitted. We agree that splitting up the sources of the extra 125 TgC yr⁻¹ is a better way to explain the matter and have made the following amendment starting at line 172, drawing the reader's attention to Figure 1(a) which clearly shows the difference in the treatment of monoterpenes between the mechanisms.

~~In the PI, CS2 simulates an extra ~125 TgC yr⁻¹ of reactive organic carbon emissions than ST, primarily due to the chemical treatment of monoterpenes with a smaller contribution from the extra emitted species considered by CS2 (Methods).~~ In the PI, CS2 simulates an extra ~5 TgC yr⁻¹ of ROC emissions than ST due to the wider range of emitted VOCs considered by CS2 (Archer-Nicholls et al., 2021 and Methods). In addition, CS2 features an extra ~120 TgC yr⁻¹ of reactive organic carbon produced in the atmosphere in the form of 1st generation oxidation products from monoterpenes compared to ST since monoterpene oxidation in ST does not produce any chemically-active species (Fig 1(a) and Methods).

REVIEWERS' COMMENTS

Reviewer #1 (Remarks to the Author):

The authors have now addressed most of my major comments. I still feel that in addition to 2xBVOC in preindustrial, it would have been worthwhile to assess a 2xBVOC simulation in the present day. Even if the authors decided not to compare present day to preindustrial, a simulation with 2x BVOC in present day would help to assess sensitivities of forcing to change in BVOC emissions and how these sensitivities change in present day compared to preindustrial.

Reviewer #3 (Remarks to the Author):

I believe the authors' clarifications in response to previous reviews have improved the manuscript and satisfactorily addressed my concerns, and that the manuscript is ready for publication.

Response to Reviewers

Text which has been added to the manuscript is coloured red. Original manuscript text is in blue. The locations of changes are stated. We hope these revisions address the concerns of the reviewers.

Reviewer #1 (Remarks to the Author):

The authors have now addressed most of my major comments. I still feel that in addition to 2xBVOC in preindustrial, it would have been worthwhile to assess a 2xBVOC simulation in the present day. Even if the authors decided not to compared present day to preindustrial, a simulation with 2x BVOC in present day would help to assess sensitivities of forcing to change in BVOC emissions and how these sensitivities change in present day compared to preindustrial.

In response we have made the following addition (starting at line 443) to the conclusion in the section where we previously noted that the dependence of atmospheric chemical composition would mean that a doubling of BVOC emissions would likely have a different climatic impact in a present day or future climate.

The wide-ranging influence of oxidants and chemistry identified in this study, and the attendant dependence on atmospheric chemical composition, means a doubling of BVOC emissions in a present-day or future climate is likely to have a different climatic impact to that simulated here. Such experiments would provide further information regarding the sensitivity of BVOCs' climatic impact to background atmospheric conditions and make for interesting follow up studies.

We believe this makes clear to readers the next steps for this work.